# Lifted Uniform Quantization for Extreme Low-bit Large Language models

## Abstract

Pushing large language models to extreme low bit-widths (e.g., 2-bit) is a critical frontier for efficient deployment, yet it presents a daunting challenge to preserving model accuracy. Current methods are trapped in a fundamental trade-off: Vector Quantization (VQ) maintains accuracy by learning expressive codebooks but is crippled by its computationally expensive, non-parallelizable lookup operations. Conversely, Uniform Quantization (UQ) is exceptionally efficient but suffers a precipitous drop in quality at such low bit-widths. To break this impasse, we propose **Lifted Uniform Quantization (LiftUQ)**, a new paradigm that encodes weights in an expanded latent space using ultra-low-bit uniform quantization (1-bit in our practice), and then applies a trainable dimensionality reduction linear transformation to project them into the original space, forming non-uniform code-points without any look-up codebook. This lifted–projected representation recovers and even surpasses the expressive power of vector quantization while retaining the decoding efficiency of scalar uniform quantization. To make LiftUQ applicable to arbitrary layers, we further learn a whitening transform to produce approximately independent Gaussian-like channels, then apply the same lifted–projected encoding. LiftUQ marks a significant breakthrough in extreme low-bit quantization. Our experiments validate that it is the **first framework to bridge the long-standing accuracy gap between uniform and vector quantization**, consistently matching or surpassing VQ performance on Llama and Qwen models—for instance, suffering less than a 2.7/1.1-point accuracy degradation on Llama-3-70B at 2/3-bit. Critically, this high accuracy is achieved with exceptional efficiency, boosting throughput up to $6.7\times$ over FP16 by combining the inherent speed of uniform decoding with a lightweight linear projection. This establishes LiftUQ a new, superior paradigm for practical quantization.

## 1 Introduction

Large language models (LLMs) (Touvron et al., 2023; Bai & et al., 2023; Dubey & et al., 2024; Touvron & et al., 2023; DeepSeek-AI, 2024) have become a cornerstone of modern AI, delivering state-of-the-art performance in complex reasoning and generation tasks. However, this progress is enabled by massive parameter counts, which impose substantial deployment challenges: models can require massive storage and suffer significant latency bottlenecks in owing to frequent off-chip memory accesses.

Weight-only quantization has emerged as an effective strategy to address these challenges. For example, reducing weights to 4-bit precision reduces model size by approximately a factor of four and proportionally reduce memory access overhead. Furthermore, advanced quantization optimization techniques effectively mitigate the accuracy degradation typically induced by low-precision representation. In particular, state-of-the-art uniform quantization (UQ) (Frantar et al., 2022; Ashkboos et al., 2024; Chen et al., 2024) achieves negligible accuracy loss at 4-bit precision by employing fine-grained quantization groups (Tseng et al., 2024a; Egiazarian et al., 2024; Liu et al., 2024a) and channel-wise transformations (e.g., scaling, orthogonal rotations). These operations incur minimal computational overhead, making UQ highly efficient in practice. However, UQ exhibits substantial performance degradation at ultra-low precisions (e.g., 2-bit or below).

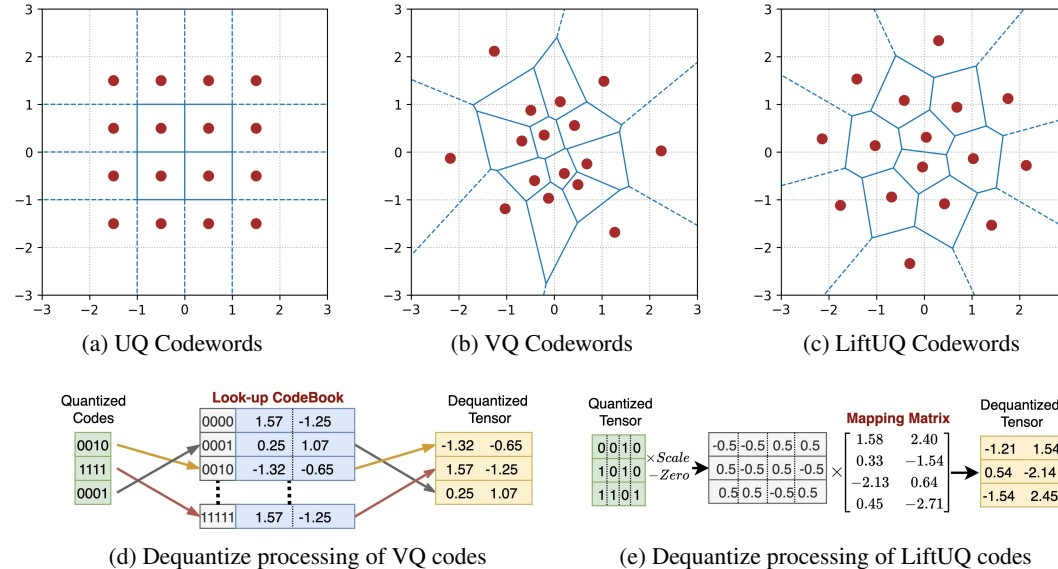

(a) UQ Codewords  (b) VQ Codewords  (c) LiftUQ Codewords

(d) Dequantize processing of VQ codes  (e) Dequantize processing of LiftUQ codes

Figure 1: **Architectural Comparison of UQ, VQ, and LiftUQ.** Subplots (a-c) visualize the 2D codeword distributions for the three methods. Uniform Quantization (a) forms a rigid, grid-like lattice. Vector Quantization (b) learns unstructured centroids that adapt to the data distribution. In contrast, LiftUQ (c) generates a structured yet non-uniform codebook, inheriting properties from both. Subplots (d-e) illustrate the critical difference in their dequantization pipelines. While VQ (d) relies on a memory-intensive and hardware-unfriendly lookup table (LUT), LiftUQ (e) employs a computationally efficient linear transformation (a simple matrix-vector product). This fundamental architectural advantage allows LiftUQ to achieve the expressive power of non-uniform quantization without the significant inference overhead of VQ.

Departing entirely from scalar methods, vector quantization (VQ) offers a more accurate alternative under such constraints. By encoding a weight vector $w \in \mathbb{R}^d$ into one of $2^{d \cdot b}$ codewords in a learned codebook (where $b$ is the per entry bitwidth), VQ captures inter-channel correlations and achieves denser coverage of the representation space. Larger vector dimensions further strengthen this effect, enabling VQ to substantially outperform UQ in ultra-low-bit regimes. Nevertheless, these gains come at the cost of expensive decoding: the required codebook size $d \cdot 2^{d \cdot b}$ is exponentially in $d \cdot b$, creating a prohibitive cache consumption. Codebook lookups also induce irregular and inherently sequential memory accesses. Consequently, VQ decoding is far less efficient than the fully parallel matrix or vector operations leveraged by UQ. Recent research reports the decoding throughput of VQ models to be highly unstable, under some circumstances even slower than the full precision model (Liu et al., 2024a). Achieving practical performance with VQ thus relies on extensive, platform-specific operator optimizations, adding a significant engineering burden.

To resolve this fundamental accuracy-efficiency trade-off, we propose Lifted Uniform Quantization (LiftUQ), a new paradigm that combines the advantages of both UQ and VQ. The key insight behind LiftUQ is that a non-uniform quantization codewords distribution, as achieved in VQ, can be estimated with the linear projection of a set of simple UQ codewords defined in a higher-dimensional space (Figure 1). This effect resembles observing a dense three-dimensional lattice of points from a two-dimensional perspective, where the projected density appears higher at the center—mirroring a Gaussian-like distribution (see Figure 2).

Concretely, LiftUQ represents weight vectors as learned linear projections from a simple, uniform 1-bit lattice in a higher-dimensional "**lifted**" space. To make this approach broadly applicable, we introduce a learnable lightweight whitening transform that reshapes weights to be more amenable to this projection, which can be fused into a single, efficient linear mapping at inference. The efficient lookup-table-free decoding architecture and the non-uniform codewords generation mechanism form the basis of our contributions.

Our main contributions are threefold:

- **A Novel Quantization Framework.** We introduce LiftUQ, which generates highly expressive, structured non-uniform codebooks from an efficient uniform foundation, effectively unifying the strengths of UQ and VQ.

- **A Hardware-Friendly Decoding Architecture.** We replace the memory-intensive lookup-table (LUT) bottleneck of VQ with a simple, computationally efficient linear transformation, making it significantly more suitable for GPU acceleration.

- **State-of-the-Art Performance.** Through extensive experiments, we demonstrate that LiftUQ establishes a new state of the art, achieving the accuracy of leading VQ methods with computational efficiency approaching that of UQ.

## 2 RELATED WORK

Weight-only quantization has emerged as one of the most effective strategies for deploying large language models (LLMs) under strict memory and latency constraints.

**Uniform scalar quantization (UQ)** is the most widely used approach, where a floating-point weight vector $w$ is represented as $w_q \cdot s$, with $w_q$ storing low-bit integer calues and $s$ is a floating scaling factor. Due to the non-uniform value distribution of LLM weights, recent UQ methods introduce lightweight preprocessing to make weights more amenable to quantization. For example, some works group channels according to activation energy and apply group-wise quantization, prioritizing the preservation of important channels (e.g., AWQ (Lin et al., 2024), BiLLM (Huang et al., 2024)). When the specially treated weights are interpreted as a low-rank branch, these methods can be adapted for quantization-error compensation using low-rank adaptation techniques (LoRA), as in QLoRA (Dettmers et al., 2023) and FBQuant (Liu et al., 2025). Other works apply importance-aware scaling to reduce quantization errors on sensitive weights (e.g., AWQ, SmoothQuant (Xiao et al., 2022), OmniQuant (Shao et al., 2023), OSTQuant (Hu et al., 2025)). An alternative line of research focuses on reshaping weight distributions to be more amenable to quantization prior to UQ. Matrix-based transforms can make weight distributions more uniform and mitigate the impact of outliers (e.g., QuIP#, QuIP (Chee et al., 2023), Quarot, SpinQuant (Liu et al., 2024b), AffineQuant(Ma et al., 2024), FlatQuant(Sun et al., 2024)).

**Non-uniform scalar quantization** methods have improved performance by creating specialized, non-uniform levels for individual weights. These approaches range from using data-type formats (e.g., FP4 (Liu et al., 2023)), to leveraging data distribution quantiles (e.g., NF4 (Dettmers et al., 2023)), or constructing levels via additive combinations of learned basis values (e.g., BCQ(Xu et al., 2018; Park et al., 2025)). However, by operating on scalars, they inherently miss the opportunity to model inter-dimensional correlations.

**Vector quantization (VQ)** compresses high-dimensional weight vectors by mapping each to its nearest representative vector (codeword) from a finite, learned codebook $\mathbf{K}$. Decoding is given by $w = \mathbf{K}[w_q] \cdot s$, where $w_q$ stores codeword indices. Compared to UQ, VQ exploits inter-element correlations and better fits non-uniform distributions, offering superior accuracy in ultra-low-bit regimes. However, VQ decoding is less hardware-friendly: the codebook size scales as $d \cdot 2^{d \cdot b}$, where $d$ is the vector dimension and $b$ bitwidth per entry, which imposes a large cache footprint, and the required codebook lookups introduce irregular, sequential memory accesses. To address these issues, recent works have focused on efficient codebook designs, such as additive codebooks that decompose a vector into the sum of smaller codebooks (Egiazarian et al., 2024), lattice-based quantization with compact representations (Tseng et al., 2024a). In addition, techniques proven effective in UQ—such as linear transforms for distribution shaping or importance-based quantization grouping (Liu et al., 2024a)—have also been integrated into VQ frameworks for advanced accuracy.

## 3 LIFTED UNIFORM QUANTIZATION FOR LLMS

### 3.1 MOTIVATION

While highly efficient, uniform quantization (UQ) is fundamentally mismatched with the non-uniform distribution of LLM weights. Even after applying whitening transforms — which reshape weight distributions to be approximately independent and identically distributed (i.i.d.) Gaussian

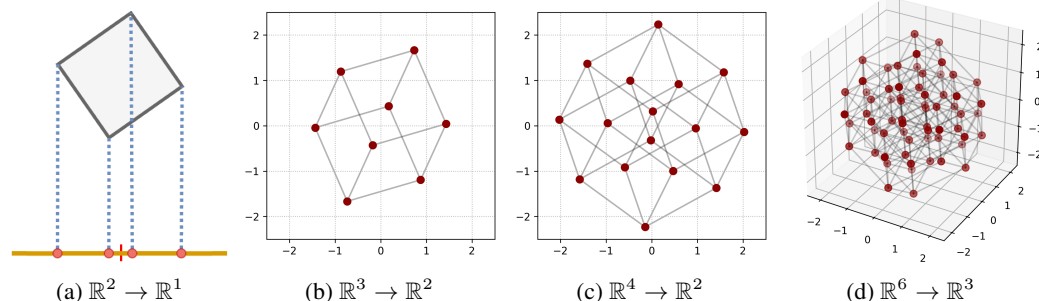

(a) $\mathbb{R}^2 \to \mathbb{R}^1$      (b) $\mathbb{R}^3 \to \mathbb{R}^2$      (c) $\mathbb{R}^4 \to \mathbb{R}^2$      (d) $\mathbb{R}^6 \to \mathbb{R}^3$

Figure 2: Visualization of Codewords Generation in Lifted Uniform Quantization. Our method generates a structured, non-uniform codebook by projecting a simple, uniform lattice from a high-dimensional "lifted" space onto a lower-dimensional target subspace. Specifically, the vertices of a $d_s \cdot b$ -dimensional hypercube (e.g., $\{+1, -1\}^{d_s \cdot b}$) are projected via a learned transformation $\boldsymbol{M}$. This process effectively combines the structural simplicity of uniform quantization with the high representational power of vector quantization. The subplots illustrate the resulting learned codewords for different projection dimensionalities.

— a uniform grid remains a sub-optimal choice. While optimal non-uniform quantizers like Vector Quantization (VQ) or data-aware codebooks (e.g., NF4) exist, they rely on expensive lookup tables (LUTs), creating an intractable accuracy-efficiency trade-off. This forces a choice between a fast-but-inaccurate model (UQ) and an accurate-but-slow one (VQ).

Our key insight is that an expressive, non-uniform codebook can be procedurally generated without a LUT. We achieve this by first representing quantization indices on a simple, uniform grid in a higher-dimensional "**lifted**" space, and then using a learned linear projection to map these points into the target weight space. As visualized in Figure 2, this projection transforms a simple hyper-cubic lattice into a structured, non-uniform codebook tailored to the Gaussian distribution. This "lift-then-project" approach, which forms the core of our LiftUQ framework, achieves the expressive power of VQ while leveraging only efficient, hardware-friendly linear operations.

Therefore, our LiftedUQ method is composed of three core phases. In Section 3.2, we learn a projection matrix $\boldsymbol{M}$ that optimally maps a uniform grid in a high-dimensional space to a non-uniform grid tailored for an i.i.d. Gaussian distribution. In Section 3.3, we learn a lightweight layer-wise whitening transformation $\boldsymbol{D}$ to convert weight distribution to i.i.d. Gaussian. Finally, in Section 3.4, we quantize the whitened weights with the codewords generated by $\boldsymbol{M}$ so we can perform an efficient UQ decoding: $\boldsymbol{o} = \mathrm{diag}(\boldsymbol{s})\boldsymbol{W}_q(\boldsymbol{M}\boldsymbol{D}^*\boldsymbol{a}^T)$.

### 3.2 PHASE 1: TRANSFORMATION FROM LIFTED UNIFORM GRID TO SUBSPACE LATTICE

The first phase of LiftedUQ learns a transformation matrix $\boldsymbol{M} \in \mathbb{R}^{d_s \times (d_s \cdot b)}$ that maps a lifted uniform grid in a high-dimensional space to a $d_s$-dimensional vector. We refer to $d_s$ as the subspace dimension, as it defines the dimensionality of the vector space in which the reconstructed weights reside. The optimization problem to solve $\boldsymbol{M}$ is defined as:

$$\boldsymbol{M}^* = \arg\min_{\boldsymbol{M}} \mathbb{E}_{\boldsymbol{w} \sim \mathcal{W}} \left[ \min_{\boldsymbol{y} \in \{-1, +1\}^{d_s \cdot b}} \left\| \boldsymbol{w} - \boldsymbol{M}\boldsymbol{y} \right\| \right], \tag{1}$$

where $\mathcal{W}$ denotes the target weight distribution (approximated by a Gaussian during training), and $\mathbf{y}$ indexes points from the lifted uniform grid. This implies that the nearest-neighbor rounding operation $\min \|\boldsymbol{w} - \boldsymbol{M}\boldsymbol{y}\|$ cannot be decoupled into independent scalar roundings, and thus exact decoding requires enumeration of all candidate $\boldsymbol{y}$. Because the nearest-neighbor operator is non-differentiable, we employ a differentiable **softmin** approximation during training to enable back-propagation. We obtain $\boldsymbol{M}$ via gradient-based optimization. In each iteration, we generate 1000 random Gaussian samples and minimize their reconstruction error against the nearest grid points.

We find that a larger subspace dimension $d_s$ systematically improves the encoding quality for a Gaussian source, as it allows for a richer set of reconstruction vectors. However, this comes at the cost of significantly increased training time, as shown in Table 1.

Table 1: Trade-off analysis for $M$ matrix dimensions in 2-bit quantization (b=2). Training time is for learning a single $M$ matrix. The Shannon limit represents the theoretical minimum MSE for a Gaussian source.

| Method | Int2 | LiftedUQ | | | | Shannon |
|---|---|---|---|---|---|---|
| $(d_s \times d_r)$ | | 4x8 | 8x16 | 10x20 | 12x24 | Limit |
| MSE (Gaussian, ↓) | 0.119 | 0.0998 | 0.0904 | 0.0875 | 0.0867 | 0.0625 |
| Training Time (↓) | - | 1 min | 10 min | 30 min | 1.5 h | - |
| Lifting Time / 1M Params (↓) | - | ≪1s | 1s | 18s | 5.5 min | - |

The exponential growth in search space makes training a layer-specific $M$ computationally infeasible. For example, using an exhaustive search over all $y \in \{+1, -1\}^{d_s \cdot b}$, training a single 10x20 matrix takes 30 minutes. Applying this to each linear layer of a 7B model would extend the quantization time to an impractical 100+ hours. Even with heuristic methods to prune the search space (see Appendix Y for details), the exponential nature of the problem persists. To circumvent this, we instead train a single, globally optimal transformation $M$ on a standard $d_s$-dimensional Gaussian distribution and reuse it across all layers after applying the whitening process (Section 3.3).

For our main experiments, we use moderate dimensions, setting $M \in \mathbb{R}^{20 \times 10}$ for 2-bit quantization and $M \in \mathbb{R}^{18 \times 6}$ for 3-bit. Since the projection matrix $M$ can be pre-fused with the whitening transformation, its complexity (determined by the subspace dimension $d_s$) introduces no additional computational cost during inference.

A key advantage of LiftedUQ is its natural support for **fractional bitwidths**. Since representational capacity is encoded along the lifted channel dimension $d_r = d_s \cdot b$ rather than by a fixed scalar bitwidth, intermediate configurations such as $M \in \mathbb{R}^{22 \times 10}$ (2.2-bit) or $M \in \mathbb{R}^{25 \times 10}$ (2.5-bit) are possible. This flexibility allows for fine-grained control over the performance-memory trade-off at deployment. For instance, it enables deploying a 70B model with 2.5-bit quantization on a single 24GB GPU—a feat infeasible with conventional uniform quantization schemes.

### 3.3 PHASE 2: LEARNED WHITENING TRANSFORMATION FOR EACH LAYER

In the second phase, LiftedUQ learns a lightweight whitening transformation $D$ for each linear layer. While prior works have employed linear transformations to improve quantization robustness—such as scaling, rotations, or affine mappings — our whitening transform $D$ is explicitly designed to reshape layer weights into an approximately i.i.d. Gaussian distributions, making them directly compatible with the LiftedUQ lattice obtained in Section 3.2.

To achieve both efficiency and representational power, we parameterize $D$ in a decomposed form:

$$D = \text{diag}(s_1)(P_1 \otimes P_2)\text{diag}(s_2) \tag{2}$$

where activation multiplication by $D^{-1}$ scales as $\mathcal{O}(n\sqrt{n})$, significantly lower than the $\mathcal{O}(n^2)$ cost of dense matrix multiplication in $Wa^T$. $n$ is the input dimension.

In this structure, $s_1$ and $s_2$ are diagonal matrices, performing lightweight per-channel rescaling; $P_1$ and $P_2$ are $\sqrt{n} \times \sqrt{n}$ matrices whose Kronecker product provides channel intermixing and whitening ability. This design offers both computational efficiency and functional expressivity.

Specifically: (1) First Scaling $s_1$ redistributes quantization error according to channelwise activation magnitudes. Inspired by AWQ, channels with larger activations are down-scaled to reduce their relative quantization error. To avoid invalidating the assumption of approximately constant quantization noise energy, we initialize $s_1$ using the relative activation variances and apply truncation to mitigate extreme outliers. (2) Interleaved whitening $P_1 \otimes P_2$ mixes channels to locally approximate i.i.d. Gaussian structure. We initialize $P_1$ and $P_2$ as orthogonal (via Hadamard matrices or truncated–orthogonalized variants when dimension mismatch occurs), so that channel energy is preserved, and outliers are diffused across dimensions as in QuIP. During training, no orthogonality

constraint is enforced, allowing richer adaptation capacity. (3) Final Shaping $s_2$ further refines per-channel variance normalization, ensuring stronger isotropy with respect to the LiftedUQ lattice. In addition, by introducing an additional degree of freedom in rescaling, $s_2$ expands the optimization space, which will be exploited in Section 3.4 during the joint arithmetic fusion with the mapping matrix $M$.

Crucially, $D$ is invertible by construction, ensuring that whitening and de-whitening form a reversible process. This reduces overfitting risk since intrinsic weight information is preserved. The optimization objective thus becomes:

$$\underset{s_1, P_1, P_2, s_2}{\arg\min} \; \mathcal{L}\left(W a^T - \text{Quant}_{ste}\left(W D\right) D^{-1} a^T\right),\tag{3}$$

where quantization noise is minimized under the reversible transformation. In practice, this optimization is performed block-wise using standard optimizers (e.g., Adam).

This decomposed whitening transform achieves two objectives simultaneously: (i) efficient allocation of quantization error across activation-sensitive channels, and (ii) reshaping channel distributions to well-approximated i.i.d. Gaussians, thereby enabling the effective application of Phase 1 LiftedUQ grids at negligible computational and storage overhead.

## 3.4 PHASE 3: LATTICE QUANTIZATION AND INTRA-BLOCK CORRECTION

In the final phase, LiftedUQ integrates the learned transformations and refines model performance through block-wise fine-tuning. Having obtained a projection matrix $M$ from Phase 1 and a whitening transform $D$ from Phase 2, we first quantize the whitened weights. The quantization and reconstruction process can be formally expressed as:

(**i**) Whitening and Standardization: The layer weights $W$ are first whitened and standardized to ensure compatibility with the trained LiftedUQ lattice:

$$\underset{OC \times IC}{W'} = \text{diag}(\text{std}(W D)^{-1}) \underset{OC \times 1}{} \underset{OC \times IC}{(W D)}.\tag{4}$$

(**ii**) Lattice Quantization: For the purpose of quantization, we view the elements of $W'$ as a sequence of $C$ blocks, where each block is a $d_s$-dimensional vector and $C = \lceil \frac{OC \cdot IC}{d_s} \rceil$. The standardized weights $W$ are quantized by finding the nearest neighbor in the LiftedUQ lattice, yielding a low-bit representation $W_q \in \{-1, +1\}^{C \times (d_s \cdot b)}$:

$$W_q = \underset{\hat{W}_q \in \{-1, +1\}^{C \times (d_s \cdot b)}}{\arg\min} \left\| \underset{C \times d_s}{W'} - \hat{W}_q M^T \right\|_F^2, C = \lceil \frac{OC * IC}{d_s} \rceil.\tag{5}$$

Here, $\underset{C \times d_s}{W'}$ denotes the matrix $W'$ after being reshaped into a $C \times d_s$ layout to align with $M$.

(**iii**) Reconstruction: The layer output can be directly computed by reconstructing the weights and multiplying with the input activations $a$:

$$o = \text{diag}(s) \cdot \underset{OC \times IC \cdot b}{W_q} D^* a^T\tag{6}$$

where $s = \text{std}(W D)$ and $D^* = M^T D^{-1}$. For this to be computationally advantageous, we structure $D$ such that $M$ can be merged with sub-components $P_2^{-1}$. By enforcing a constraint in Phase 2 that $s_2$ remains constant within each block processed by $\tilde{M}$.

This formulation is equivalent to a 1-bit uniform quantization scheme, where a low-bit matrix $W_q$ is down-projected via $D^*$ before matrix multiplication with activation $a$. The order of operations can be dynamically chosen to optimize latency; for instance, computing $(D^* a^T)$ first is highly efficient during decoding as $a$ has mini batchsize.

Finally, to recover performance lost during quantization, we perform block-wise fine-tuning on both the low-bit representation $W_q$ and the lightweight transformation matrix $D$. Using the Adam optimizer, we minimize the reconstruction loss over a small calibration dataset for each block:

$$\underset{W_q, D^*}{\min} \; \mathbb{E}_{a \sim \mathcal{D}_{\text{calib}}} \left\| \mathbb{F}(W_{fp} a^T) - \mathcal{F}(W_q D^* a^T) \right\|_F^2,\tag{7}$$

where $\mathcal{D}_{\text{calib}}$ is the calibration data and $\mathbb{F}$ is the transformer block. This local adaptation step is critical for achieving near-lossless quantization performance. Further training details are provided in Appendix A.

### 3.5 FAST AND FLEXIBLE DECODING

A key advantage of LiftedUQ is its highly efficient and flexible decoding architecture. The combined whitening and projection transform, denoted as a single matrix $D$, can be dynamically applied based on the inference workload, ensuring minimal overhead. There are two primary modes of operation:

1. **Apply to Activations First ($W_q(DA)$):** This approach is ideal for memory-bound scenarios such as autoregressive decoding with small batch sizes, as it avoids materializing the full-precision dequantized weights.
2. **Apply to Weights First ($(W_qD)A$):** This mode is better suited for compute-bound scenarios like large-batch prefilling, where the one-time cost of dequantizing the weights is amortized over a large number of input tokens.

Table 7 provides a formal breakdown of the asymptotic computational and storage costs per layer, confirming the efficiency of both modes. Considering a typical 8192×8192 layer with 2-bit quantization in a $W_q(DA)$ setting, the additional FLOPs constitute a mere **3.3%** overhead relative to the main GEMM operation, while the parameter storage adds only a **0.6%** overhead.

## 4 EXPERIMENTS

We present a comprehensive evaluation of Lifted Uniform Quantization (LiftUQ) to demonstrate its advantages in compression quality, inference efficiency, and flexibility. In Section 4.1, we show that LiftUQ outperforms state-of-the-art uniform (UQ) and vector quantization (VQ) methods, particularly in the most challenging 2- to 3-bit weight-only regime. We highlight LiftUQ's native support for fractional bit-widths in Section 4.2, which enables a Pareto-optimal trade-off between model size and performance. Section 4.3, we validate the inference efficiency of LiftUQ, demonstrating superior decoding throughput compared to VQ-based approaches. Finally, we discuss the limmitation of our method in Section 4.5.

### 4.1 MAIN RESULTS ON COMPRESSION QUALITY

**Experimental Setup**. We evaluate LiftUQ on the Llama-2 and Llama-3 families, spanning five different model sizes, to demonstrate its broad applicability. Our evaluation focuses on the ultra-low 2-bit and 3-bit weight-only quantization regimes. We report perplexity (PPL) on the WikiText-2(Merity et al., 2016) and C4(Raffel et al., 2020) validation sets with a context length of 2048. Additionally, we assess zero-shot accuracy on five common-sense reasoning benchmarks: ARC-c, ARC-e(Clark et al., 2018), HellaSwag(Zellers et al., 2019), PIQA(Bisk et al., 2020), and Wino-Grande(Sakaguchi et al., 2021).

**Post-quantization Fine-tuning**. Post-quantization fine-tuning has emerged as a highly effective technique for maximizing the performance of low-bit models. Its efficacy is demonstrated by its adoption across top-performing methods, including, EfficientQAT (EQAT), QuIP#, AQLM, and VPTQ. This paradigm strikes an optimal balance between the simplicity of Post-Training Quantization (PTQ) and the high performance of Quantization-Aware Training (QAT), as it only requires fine-tuning quantization-related parameters (e.g., scales, transformations) on a small calibration dataset (1-16M tokens). To unlock the full potential of our method, we adopt this protocol for LiftUQ. Consequently, all results presented for LiftUQ and the baselines reflect the performance after applying this fine-tuning step, unless specified otherwise. Further details are provided in the Appendix B. And the sensitivity of our method to the calibration data is discussed in Appendix C.

**Main Results**. Table 2 presents the PPL results on WikiText-2 and C4, while Table 3 summarizes the zero-shot accuracy for 2-bit quantization across all models (3-bit results are in the appendix). In these results, LiftUQ demonstrates a substantial performance gap over leading uniform quantization (UQ) methods. Even when operating at a coarser per-channel granularity, LiftUQ significantly outperforms group-wise (g64) methods like OmniQ and EQAT. For instance, on the Llama3-70B model,

LiftUQ improves the average PPL by 0.74 and average accuracy by 4.71% over EQATg64. Critically, this superior performance is achieved with an 11% smaller model footprint with per-channel quantization, highlighting the efficiency of our lifted encoding scheme. When compared against state-of-the-art vector quantization (VQ) methods, which are renowned for their high compression quality, LiftUQ consistently achieves a slight yet noticeable advantage. While QuIP#, AQLM, and VPTQ exhibit competitive and comparable performance after fine-tuning, LiftUQ surpasses them across nearly all models and evaluation metrics, establishing a new state of the art in ultra-low-bit weight quantization. Furthermore, LiftUQ outperforms the prior work PTQ1.61 (Zhao et al., 2025) in the 1.58-bit setting in Table 8. We also provide an wider experimental evaluation in Appendix I.

Table 2: Llama-2 and Llama-3 perplexity ($\downarrow$) on Wikitext2 and C4, context length 2048.

| Method | Type | Bits | 2-7 W2 | 2-7 C4 | 2-13 W2 | 2-13 C4 | 2-70 W2 | 2-70 C4 | 3-8 W2 | 3-8 C4 | 3-70 W2 | 3-70 C4 |
|---|---|---|---|---|---|---|---|---|---|---|---|---|
| FP16 | - | - | 5.47 | 6.97 | 4.88 | 6.47 | 3.32 | 5.52 | 6.14 | 8.88 | 2.85 | 6.73 |
| GPTQ | UQ | 2.00 | NaN | NaN | Inf | Inf | 25.30 | 48.82 | Inf | - | 11.90 | - |
| GPTQ-g128 | UQ | 2.13 | 50.75 | 36.76 | 43.84 | 23.07 | NaN | NaN | - | - | - | - |
| Quarot | UQ | 2.00 | 22.07 | - | 10.41 | - | 5.60 | - | - | - | - | - |
| OmniQ-g64 | UQ | 2.25 | 9.62 | 12.72 | 7.56 | 10.05 | 6.11 | 7.68 | - | - | - | - |
| EQAT-g64 | UQ | 2.25 | 6.86 | 8.50 | 5.96 | 7.59 | 4.52 | 6.38 | 9.41 | 12.77 | 6.07 | 9.23 |
| LiftUQ-noFT | UQ | 2.02 | 6.97 | 8.53 | 5.90 | 5.74 | 4.24 | 6.19 | 9.60 | 13.12 | 5.85 | 8.82 |
| LiftUQ | UQ | 2.02 | 6.58 | **8.21** | **5.66** | **7.35** | **4.13** | **6.09** | 8.61 | **11.97** | **5.31** | **8.51** |
| AQLM-noFT | VQ | 1.97-2.07 | 7.24 | 8.96 | 6.06 | 7.80 | 4.49 | 6.36 | - | - | - | - |
| AQLM | VQ | 1.97-2.07 | 6.61 | 8.28 | 5.72 | 7.44 | 4.19 | 6.13 | - | - | - | - |
| QuIP# | VQ | 2.00 | 6.66 | 8.35 | 5.74 | 7.45 | 4.16 | 6.12 | - | - | - | - |
| VPTQ | VQ | 2.02-2.08 | **6.57** | 8.27 | 5.69 | 7.41 | 4.17 | 6.13 | 9.29 | - | 5.60 | 8.82 |
| GPTQ | UQ | 3.00 | 8.37 | 9.81 | 6.44 | 8.02 | 4.82 | 6.57 | - | - | - | - |
| GPTQ-g128 | UQ | 3.13 | 6.29 | 7.89 | 5.42 | 7.00 | 3.85 | 5.85 | 9.58 | 11.66 | 5.25 | 8.64 |
| Quarot | UQ | 3.00 | 6.09 | - | 5.37 | - | 3.72 | - | - | - | - | - |
| EQAT-g128 | UQ | 3.13 | 5.81 | 7.34 | 5.12 | 6.73 | 3.61 | 5.71 | 7.09 | 10.06 | 4.19 | 7.43 |
| UniQ | NUQ | - | - | - | - | - | - | - | 6.95 | - | 4.24 | - |
| LiftUQ | UQ | 3.02 | **5.75** | **7.31** | **5.09** | **6.71** | **3.35** | **5.67** | 6.94 | 9.96 | 3.83 | 7.34 |
| QuIP# | VQ | 3.00 | 5.79 | 7.32 | 5.10 | 6.72 | 3.56 | 5.67 | - | - | - | - |
| VPTQ# | VQ | 3.01-3.03 | 5.82 | 7.33 | 5.12 | 6.70 | 3.55 | 5.67 | 6.97 | 10.11 | **3.81** | - |

Table 3: Llama-2 and Llama-3 accuracy($\uparrow$) on 2-bit quantization.

| Model | Method | type | bits | ArcC | ArcE | HellaSwag | PiQA | WinoGrande | Avg.Acc |
|---|---|---|---|---|---|---|---|---|---|
| | FP16 | - | - | 43.52 | 76.26 | 57.16 | 78.07 | 69.22 | 64.85 |
| | AutoRound-g128 | UQ | 2.13 | 32.25 | 65.99 | 40.28 | 72.96 | 61.01 | 54.50 |
| 2-7 | EQAT-g64 | UQ | 2.25 | 36.86 | 70.96 | 51.58 | 75.30 | 65.98 | 60.14 |
| | QuIP# | VQ | 2.00 | **37.88** | **71.84** | 50.84 | 74.16 | 65.67 | 60.61 |
| | VPTQ | VQ | 2.02 | 36.95 | 69.53 | 50.33 | 74.32 | 65.04 | 59.23 |
| | LiftUQ | UQ | 2.02 | 37.46 | 70.41 | **53.23** | **75.57** | **66.85** | **60.70** |
| | FP16 | - | - | 48.29 | 79.42 | 60.07 | 79.05 | 72.22 | 67.81 |
| | AutoRound-g128 | UQ | 2.13 | 38.57 | 71.17 | 53.35 | 76.17 | 64.33 | 60.72 |
| 2-13 | EQAT-g64 | UQ | 2.25 | 41.89 | 74.83 | 55.27 | 77.04 | 68.36 | 63.48 |
| | QuIP# | VQ | 2.00 | 42.92 | 75.72 | 56.53 | 77.97 | 69.06 | 64.44 |
| | VPTQ | VQ | 2.02 | 44.03 | **76.94** | 56.76 | **78.13** | 68.27 | 64.82 |
| | LiftUQ | UQ | 2.02 | 43.69 | 76.30 | **57.09** | 77.91 | **70.01** | **65.00** |
| | FP16 | - | - | 54.44 | 82.70 | 64.77 | 82.15 | 77.98 | 72.41 |
| | AutoRound-g128 | UQ | 2.13 | 46.59 | 78.37 | 59.65 | 79.00 | 74.90 | 67.70 |
| 2-70 | EQAT-g64 | UQ | 2.26 | 50.77 | 80.13 | 61.78 | 80.14 | 74.59 | 69.48 |
| | QuIP# | VQ | 2.00 | **52.65** | **81.90** | 62.86 | **81.39** | 75.77 | **70.91** |
| | VPTQ | VQ | 2.02 | 47.70 | 77.10 | **62.98** | 77.10 | 80.3 | 74.98 |
| | LiftUQ | UQ | 2.02 | 50.94 | 80.51 | 61.83 | 80.52 | **77.43** | 70.25 |
| | FP16 | - | - | 50.43 | 80.09 | 60.17 | 79.60 | 72.61 | 68.58 |
| 3-8 | EQAT-g64 | UQ | 2.25 | 37.03 | 71.17 | 51.86 | 76.03 | 67.72 | 60.76 |
| | VPTQ | VQ | 2.07 | 36.91 | 71.03 | 52.12 | 75.12 | 65.92 | 60.22 |
| | LiftUQ | UQ | 2.02 | **40.87** | **74.33** | **53.87** | **76.55** | **68.03** | **62.73** |
| | FP16 | - | - | 60.41 | 86.99 | 66.36 | 82.37 | 80.51 | 75.33 |
| 3-70 | EQAT-g64 | UQ | 2.25 | 49.06 | 77.40 | 61.60 | 77.37 | 74.03 | 67.89 |
| | VPTQ | VQ | 2.02 | 52.65 | 81.86 | 61.71 | 80.36 | 77.90 | 70.90 |
| | LiftUQ | UQ | 2.02 | **56.14** | **84.30** | **62.31** | **81.72** | **78.53** | **72.60** |

## 4.2 FRACTIONAL BIT-WIDTHS AND THE PARETO FRONTIER

A key advantage of LiftUQ is its native ability to support fractional bit-widths. This stems from its design of encoding information in the dimensionality of the lifted space rather than rigidly in the bit-width of the quantized elements. For instance, by setting the dimensionality expansion factor to $\frac{16}{7}/\frac{19}{7}$ alongside 1-bit base quantizers, we can construct an effective 2.3/2.7-bit representation.

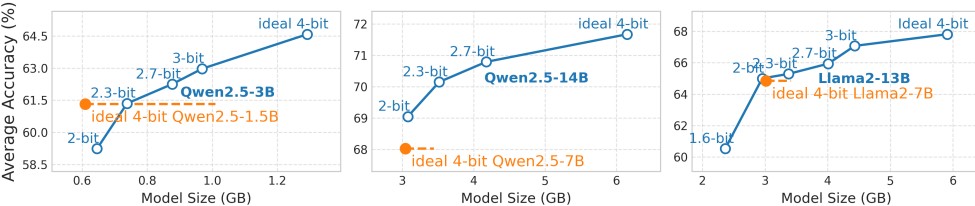

Figure 3: Fractional Bit-widths Create a New Pareto Frontier. We define the ideal 4-bit model as a 4-bit quantized model that exhibits no accuracy degradation compared to FP16.

This capability allows us to address a fundamental limitation in model deployment: LLMs are typically released in discrete, power-of-two sizes (e.g., 3B, 7B, 14B), making it difficult to find the optimal model for a specific memory budget. Following prior work (Egiazarian et al., 2024), we define a model as Pareto-optimal if it achieves the highest performance for a given storage footprint.

Our results demonstrate that LiftUQ enables larger models to dominate the Pareto frontier across a wide range of memory budgets. As shown in Table 2, the 2-bit LiftUQ version of Llama-2-13B surpasses the performance of the full-precision Llama-2-7B. Assuming a 4-bit quantization of Llama-2-7B is required to approximate its FP16 performance (a common baseline), our finding implies that the Pareto-optimal models in the 3.5 GB to 13 GB storage range are exclusively occupied by differently quantized versions of Llama-2-13B, as illustrated in Figure 3. We further validate this principle on the Qwen-2.5 series. This suggests that for achieving optimal performance under a specific memory constraint, quantizing a larger model with LiftUQ's fractional bit-widths is a more effective and significantly more economical strategy than training smaller, discrete FP16 models from scratch.

## 4.3 EFFICIENCY

Despite achieving comparable or superior compression quality, LiftUQ presents significant efficiency advantages over VQ-based methods in both training and inference. The training process for LiftUQ is markedly resource-efficient. Quantizing a 70B model requires approximately 100 hours on a single A100-80GB GPU. This computational budget is less than one-third of that reported for leading VQ methods like AQLM, substantially lowering the barrier for applying ultra-low-bit quantization.

For inference, LiftUQ's lookup-table-free architecture delivers considerable speed benefits. LiftUQ's decoding complexity is merely $\mathcal{O}(d^{1.5})$, which is less than the complexity of matrix-vector multiplication(GEMV). In our Triton-based implementation followed with BitBLAS (Wang et al., 2024) mixed-precision GEMV kernel, this approach yields up to a 6.69x throughput increase Table 5 compared to fp16 on GTX 4090D GPU. We note that this speedup is achieved without CUDA-level optimization, suggesting that further performance gains are attainable.

Vector Quantization (VQ) relies on decoding with at least $\mathcal{O}(d^2)$ complexity and a codebook that grows exponentially in size, severely limiting its scalability due to cache capacity constraints. Consequently, VQ's decoding overhead is asymptotically on par with the GEMV operation itself, creating a significant bottleneck that fundamentally lowers its maximum achievable speedup from the theoretical limit to a much smaller constant. This architectural difference results in a vast performance gap, as shown in Tables 4 and 5. Even with a minimal $2^8 \times 8$ bytes codebook, VQ's decoding time escalates rapidly with increasing matrix size. Conversely, LiftUQ's latency remains consistently low, with measured performance reaching 6.69x—nearing the 8x theoretical memory-bound speedup for 2-bit decoding. Critically, a VQ model with such a minimal codebook yields accuracy far inferior to LiftUQ.

Table 4: VQ decoding speed-up with a minimal $2^8 \times 8$ bytes codebook, bsz=1.

| ic×oc | 4096×4096 | 4096×14336 | 8192×8192 | 8192×28672 |
|---|---|---|---|---|
| Fp16 | 35.6us | 120.9us | 138.6us | 484.4us |
| Look-up | 10.3us | 40.2us | 36.8us | 123.6us |
| Full time | 19.1us | 57.9us | 56.7us | 193.7us |
| Speed-up | 1.86x | 2.09x | 2.44x | 2.50x |

Table 5: LiftUQ reaching 6.69x speed-up on 70B FFN layer at decoding stage.

| ic×oc | 4096×4096 | 4096×14336 | 8192×8192 | 8192×28672 |
|---|---|---|---|---|
| Fp16 | 35.6us | 120.9us | 138.6us | 484.4us |
| Transform | 5.3us | 5.3us | 8.4us | 8.4us |
| Full time | 14.1us | 23.0us | 28.2us | 71.8us |
| Speed-up | 2.47x | 5.26x | 4.91x | 6.69x |

### 4.4 ABLATION STUDY

To validate the effectiveness of our proposed whitening transform and the LiftedUQ framework, we conduct a series of ablation studies on the Llama-2-7B model. We first analyze the impact of the initialization strategy for the whitening transform components ($P_1$, $P_2$, $s_1$, $s_2$). Our findings indicate that a structured initialization is not merely beneficial but critical for training stability. Multiple attempts using random or identity matrices for the transformations $P_1$ and $P_2$ consistently resulted in numerical instability and training divergence. In contrast, stable convergence was reliably achieved only when initializing $P_1$ and $P_2$ with Hadamard matices.

We then quantitatively analyze the contribution of each component. The results, summarized in Table 9, demonstrate that each element provides a significant and cumulative contribution to the final performance. Furthermore, to showcase the scalability of our approach, we experimented with a larger projection matrix ( $M \in \mathbb{R}^{16 \times 32}$ for 2-bit). This configuration further reduced the perplexity to 6.50, confirming that LiftedUQ's performance can be systematically improved by increasing the subspace dimension.

### 4.5 WHERE IS THE LIMITATION OF LIFTUQ?

The primary limitation of LiftedUQ is the performance gap to the theoretical Shannon limit. Our analysis shows that LiftedUQ's achievable MSE for 2-bit quantization asymptotes to approximately 0.08 (Figure 4), whereas the Shannon limit is 0.0625. This translates to an information-theoretic gap of 0.019 to 0.07 bits, representing the potential headroom for a theoretically perfect—though perhaps undiscovered—2-bit quantizer.

However, this limitation is uniquely offset by LiftedUQ's native support for fractional bitwidths, which sidesteps the rigidity of integer-bit schemes. This flexibility is critical for achieving Pareto optimality, enabling, for instance, the deployment of a 70B model on a single 24GB GPU via 2.4-bit quantization—a feat infeasible for standard integer

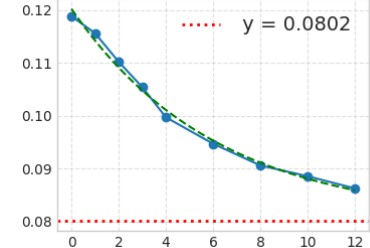

Figure 4: As the $d_s$ increases, the MSE steadily decreases, eventually approaching an asymptotic limit.

methods. Such a configuration would likely outperform even a hypothetical, Shannon-limit 2-bit quantizer. Thus, LiftedUQ is exceptionally effective at maximizing model performance within real-world hardware constraints, even if it does not reach the absolute theoretical limit for a fixed integer bitwidth. Accordingly, we discuss and compare our method against the sota vector quantizer, QTIP, in Appendix G.

## 5 CONCLUTION

In this work, we introduced Lifted Uniform Quantization, a novel framework that resolves the fundamental accuracy-efficiency trade-off in extreme low-bit LLM compression. LiftUQ bridges the gap between Uniform Quantization and Vector Quantization by representing weights as a learned linear projection from a lifted, uniform lattice, thereby achieving VQ-level accuracy without its expensive lookup-table overhead. Our extensive experiments validate that LiftUQ establishes a new state of the art, consistently matching top VQ methods while delivering up to $6.7\times$ higher throughput than FP16 execution. By replacing the bottlenecks of VQ with a hardware-friendly linear architecture, LiftUQ provides a robust and scalable foundation for the future of extreme model compression.

REPRODUCIBILITY STATEMENT

To ensure the reproducibility of our results, we have included comprehensive details of our methodology, experimental setup, and all hyperparameters in the main paper and its appendices. We will release our source code and quantized model checkpoints to facilitate verification and future work. An anonymized version of the code and checkpoints will be made available during the rebuttal period, and a public release will follow upon acceptance of the paper.

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

## A    TRAINING DETAILS FOR INTRA-BLOCK CORRECTION

This appendix details the training procedure for the block-wise correction phase described in Section 3.4. The goal of this phase is to correct for quantization errors by jointly optimizing the low-bit weights $\boldsymbol{W}_q$ and the transformation matrix $\boldsymbol{D}^*$.

Two primary strategies exist for post-quantization correction. The first, based on the Hessian matrix, involves adaptively rounding weight vectors (Frantar et al., 2022; Tseng et al., 2024a). However, this class of methods is impractical for our framework due to the prohibitive computational cost of the nearest-neighbor search required to determine the set of valid rounding candidates for each vector in our lattice.

Consequently, we adopt a more practical and effective approach: direct fine-tuning using gradient descent (Egiazarian et al., 2024; Chen et al., 2024). This method, proven viable in prior work, allows us to optimize both $\boldsymbol{D}^*$ and $\boldsymbol{W}_q$ simultaneously. Since $\boldsymbol{W}_q$ consists of discrete values, we employ the Straight-Through Estimator (STE) to approximate gradients during backpropagation.

For the correction process, we constructed a calibration dataset by randomly selecting 4,096 samples from the RedPajama dataset, with each sample having a sequence length of 2048 tokens. From this set, 128 samples were held out as a validation set. We used the Adam optimizer to minimize the Mean Squared Error (MSE) loss between the outputs of the quantized layer and the original full-precision layer. The learning rate for the transformation parameters $\boldsymbol{D}^*$ was set to $1 \times 10^{-3}$ across all models. For the $\boldsymbol{W}_q$, we used a learning rate of $2 \times 10^{-5}$ for models between 3B and 14B parameters, and a reduced rate of $1 \times 10^{-5}$ for the 70B model. The entire training process was conducted for 2 epochs.

## B    TRAINING DETAILS FOR END TO END FINE-TUNE

To further enhance model performance and globally align the quantization parameters, we perform an optional end-to-end fine-tuning step. The effectiveness of this approach for adjusting quantization parameters has been validated by several prior works (Tseng et al., 2024a; Egiazarian et al., 2024; Chen et al., 2024; Liu et al., 2024a).

This fine-tuning process optimizes the continuous parameters of our framework—specifically, the scaling parameters and the components of the transformation matrix $\boldsymbol{D}$—across all layers simultaneously. Unlike the layer-wise correction phase, this step minimizes the standard language modeling loss (i.e., Cross-Entropy) over the entire model.

For training, we used a dataset of 4,096 samples from RedPajama, each with a sequence length of 4096. We employed the Adam optimizer and trained for a single epoch. A differential learning rate scheme was applied: the learning rate for the quantization scaling parameters was set to $1 \times 10^{-5}$, while the transformation parameters used a higher rate of $3 \times 10^{-4}$. A significant advantage of this approach is its remarkable memory efficiency. Since the fine-tuning is performed on the already quantized model, the weights remain in their low-bit format throughout the process. This dramatically reduces the memory footprint, enabling us to fine-tune the entire 70B model on a single 80GB A100 GPU—a task that is infeasible for its full-precision counterpart.

## C    SENSITIVITY TO CALIBRATION DATA

To investigate the sensitivity of our fine-tuning process to the choice of calibration data, we conducted a comprehensive ablation study. We varied the calibration dataset's size, domain, and sequence length, and evaluated the impact on Llama-2-7B. For these experiments, we used a 10x20 $M$-matrix and only performed the intra-block correction (without end-to-end fine-tuning) to isolate the specific effect of the calibration data. The results are presented in Table 6.

Our findings from this study provide two key insights:

**Robustness to Data Size and Sequence Length.**    The results indicate that while performance improves as the calibration data size increases from 1M to 8M tokens, there are clear diminishing returns beyond approximately 4M tokens. Similarly, reducing the sequence length from 2048 to

Table 6: Ablation study on the calibration data for 2-bit Llama-2-7B. The default configuration used in our main experiments is highlighted in bold.

| Calibration Set | Config. (Samples $\times$ SeqLen) | WikiText-2 PPL ($\downarrow$) | C4 PPL ($\downarrow$) | Avg. 0-shot Acc. ($\uparrow$) |
|---|---|---|---|---|
| RedPajama (Small) | $512 \times 2048$ ( 1M tokens) | 7.08 | 8.66 | 60.03 |
| RedPajama (Medium) | $1024 \times 2048$ ( 2M tokens) | 7.00 | 8.59 | 60.55 |
| RedPajama (Large) | $2048 \times 2048$ ( 4M tokens) | 6.96 | 8.53 | 60.68 |
| **RedPajama (Default)** | **$4096 \times 2048$ ( 8M tokens)** | **6.97** | **8.53** | **60.70** |
| RedPajama (Short Seq) | $4096 \times 512$ ( 2M tokens) | 6.98 | 8.53 | 60.67 |
| WikiText-2 (In-Domain) | $2048 \times 2048$ ( 4M tokens) | **6.72** | 8.65 | 60.24 |

512 while keeping the total token count constant has a minimal impact on the final performance. **Our choice of 8M tokens (4096 samples $\times$ 2048 sequence length) for the main experiments was made to ensure a fair comparison with other methods, such as AQLM and EfficientQAT.**

**Impact of Domain Shift.** As expected, calibrating on a domain-matched dataset (WikiText-2) yields the best perplexity on that specific in-domain benchmark (6.72 PPL), as shown in Table 6. This specialization, however, comes at the cost of slightly degraded performance on out-of-domain benchmarks like the C4 dataset and zero-shot tasks. Using a large, general-purpose corpus like RedPajama provides a more balanced and robust performance across all evaluation metrics.

# D  DECODING OVERHEAD

Table 7: Asymptotic complexity and storage analysis per layer of size $N \times M$ at 2bit quantization.

| Method | Main GEMM FLOPs | Additional FLOPs | Weight Storage | Additional Storage |
|---|---|---|---|---|
| FP16 | $2NM$ | - | $16NM$ | - |
| **LiftUQ** (decoding) ($D\boldsymbol{A}$ first, batch=1) | $2NM$ | $O(d_s N + d_s^2 N)$ | $bNM$ | $O(d_s^2 N)$ |
| **LiftUQ** (prefill) ($\boldsymbol{W}_q D$ first, batch=$k$) | $2kNM$ | $O((d_s N + d_s^2 N)k)$ | $bNM$ | $O(d_s^2 N)$ |

Note: $k$ is batch size, $b$ is bitwidth, $d_s$ is subspace dimension.

# E  COMPARISON ON 1.58-BIT BASELINE.

Table 8: Comparison on 1.58-bit Baseline.

| Method | Type | Bits | 2-7 | | | 2-13 | | | 3-8 | | |
|---|---|---|---|---|---|---|---|---|---|---|---|
| | | | W2$\downarrow$ | C4$\downarrow$ | Avg.Acc$\uparrow$ | W2$\downarrow$ | C4$\downarrow$ | Avg.Acc$\uparrow$ | W2$\downarrow$ | C4$\downarrow$ | Avg.Acc$\uparrow$ |
| FP16 | - | - | 5.47 | 6.97 | 64.85 | 4.88 | 6.47 | 67.81 | 6.14 | 8.88 | 68.58 |
| PTQ1.61 | UQ | 1.61 | 12.70 | 17.73 | 44.14 | 9.74 | 13.64 | 49.21 | 22.90 | 33.82 | 43.99 |
| LiftUQ | UQ | **1.62** | **7.71** | **9.55** | **56.19** | **6.47** | **8.27** | **60.54** | **11.43** | **15.13** | **56.66** |

# F  ABLATION STUDY ON THE COMPONENTS OF OUR WHITENING TRANSFORM AND LIFTEDUQ FRAMEWORK

Table 9: Ablation study on the components of our whitening transform and LiftedUQ framework, evaluated on Llama-2-7B. Each column represents the addition of a new component to the configuration of the preceding column.

| Configuration | $P_1$+$P_2$ | + $a_1$ | + $a_2$ | + $10 \times 20$ M | +E2E FT | $16 \times 32$ M + E2E FT |
|---|---|---|---|---|---|---|
| WikiText-2 PPL ($\downarrow$) | 8.76 | 8.28 | 7.77 | 6.96 | 6.58 | 6.50 |

## G COMPARISON WITH QTIP

To contextualize the performance and architectural choices of LiftedUQ, we provide a detailed comparison with QTIP Tseng et al. (2024b), a state-of-the-art method in vector quantization (VQ). While both methods aim for extreme low-bit quantization, they operate under fundamentally different paradigms. QTIP advances the state-of-the-art within traditional VQ by employing Trellis Coded Quantization (TCQ) to optimize for rate-distortion performance. In contrast, LiftedUQ forges a new path by unifying the strengths of Uniform Quantization (UQ) and VQ. It employs a lift-then-project technique to deliver VQ-level accuracy and fractional bitwidth flexibility, all while preserving the simple and hardware-friendly decoding architecture of UQ.

Our comparison focuses on two critical aspects: (1) rate-distortion performance and flexibility, and (2) hardware efficiency and practical throughput.

### G.1 RATE-DISTORTION PERFORMANCE AND FLEXIBILITY

While QTIP's use of TCQ allows it to approach the Shannon limit more closely for a fixed integer bitwidth, a deeper analysis reveals LiftedUQ's unique advantages in flexibility and practical performance.

**Rate-Distortion Analysis.** Table 10 presents a rate-distortion analysis for various methods on a Gaussian source ($\sigma^2 = 1$). For strict 2-bit quantization, QTIP's MSE (0.0733) is indeed closer to the Shannon limit (0.0625) than our baseline LiftedUQ configurations. We observe a strong correlation between this theoretical MSE and the empirical model performance (Perplexity on Llama-2-7B), validating the relevance of this analysis.

Table 10: Rate-distortion analysis for quantizing a Gaussian source ($\sigma^2 = 1$) at approximately 2 bits. "Equivalent Bits" are derived from MSE via the rate-distortion function $R(D) = 0.5 \log_2(1/D)$.

| Method | Config. | MSE ($\downarrow$) | Equiv. Bits ($\uparrow$) | PPL, Llama2-7B ($\downarrow$) |
|---|---|---|---|---|
| Integer Quant. | 2-bit | 0.119 | 1.535 | - |
| QuIP# Tseng et al. (2024a) | E8 Lattice | 0.089 | 1.745 | 6.66 |
| **LiftedUQ (Ours)** | **10x20 (2.0-bit)** | **0.0873** | **1.759** | **6.59** |
| **LiftedUQ (Ours)** | **16x32 (2.0-bit)** | **0.0835** | **1.791** | **6.51** |
| QTIP Tseng et al. (2024b) | TCQ (L=16) | 0.0733 | 1.885 | 6.28 |
| **LiftedUQ (Ours)** | **15x32 (2.13-bit)** | **0.0696** | **1.922** | **6.23** |
| **LiftedUQ (Ours)** | **10x25 (2.5-bit)** | **0.0453** | **2.230** | **5.99** |
| Shannon Limit | 2-bit | 0.0625 | 2.000 | - |

**Flexibility and Pareto-Optimality.** The key advantage of LiftedUQ is its native support for **fractional bitwidths**, a capability not present in QTIP. As demonstrated in Table 10, by slightly adjusting the lifted dimension (e.g., to a 15x32 config., effective 2.13 bits), **LiftedUQ achieves an MSE of 0.0696, which not only matches but surpasses 2-bit QTIP**. This flexibility is crucial for real-world deployment. For example, when deploying a 70B model on a 24GB GPU, QTIP is constrained to a 2-bit representation. LiftedUQ, however, can be configured to use 2.5 bits to fully utilize the available memory, operating at a much more favorable point on the rate-distortion curve (MSE 0.0453) and delivering a Pareto-optimal solution that would significantly outperform 2-bit QTIP.

## G.2  HARDWARE EFFICIENCY AND PRACTICAL THROUGHPUT

A key differentiator of LiftedUQ is its inference efficiency, which stems from a fundamentally more hardware-friendly architecture.

**Architectural Comparison.**  As detailed in Table 11, LiftedUQ's decoding is activation-centric, relying on a simple, highly parallelizable matrix-vector product (GEMV-like). In contrast, QTIP's decoding is weight-centric and far more complex, involving computationally intensive transforms and non-linear decoding steps that are ill-suited for modern GPU architectures.

Table 11: Asymptotic decoding complexity comparison for a $D \times D$ weight matrix.

| Method | Core Operation | Complexity (on weights) | Hardware-Friendly? |
|---|---|---|---|
| **LiftedUQ** | Linear transforms (GEMV-like) | (Applied to activations) | **Yes** |
| QTIP | Hadamard + Non-linear codes | $O(D^2 \log D + D^2)$ | **No** |

**Empirical Throughput.**  This architectural difference translates directly into a massive performance advantage. We configured LiftedUQ to a 2.13-bit setup for a fair accuracy comparison with 2-bit QTIP. As shown in Table 12, the results are striking: even on a **less powerful consumer-grade GPU**, LiftedUQ achieves **12% higher throughput**. This is particularly significant as QTIP relies on heavily optimized custom CUDA kernels, whereas our LiftedUQ implementation uses a simple PyTorch and BitBLAS backend. This highlights not only LiftedUQ's superior performance but also its ease of deployment and platform-agnostic efficiency.

Table 12: End-to-end throughput (tokens/sec) on Llama-2-70B (batch size = 1).

| Method (Bitwidth) | Device | Throughput (tok/s) |
|---|---|---|
| QTIP ( 2.0-bit) | RTX 6000 Ada | 23.5 |
| **LiftedUQ ( 2.13-bit)** | RTX 4090D | **26.4** |

*Note: QTIP data is from their official repository. Our result is on an RTX 4090D ( 20% less compute).*

In summary, while we acknowledge QTIP's excellent theoretical compression, we argue that LiftedUQ offers a more practical and compelling solution for real-world LLM deployment. It achieves competitive or superior accuracy through its flexible fractional bitwidths, while delivering significantly higher inference throughput due to its hardware-native, UQ-based architecture. This unique combination of accuracy, efficiency, and flexibility positions LiftedUQ as a powerful and practical paradigm for extreme low-bit quantization.

## H  FULL QUANTIZATION RESULT.

Table 13: Llama-2 and Llama-3 accuracy(↑) on 3-bit quantization.

| Model | Method | type | bits | ArcC | ArcE | HellaSwag | PiQA | WinoGrande | Avg.Acc |
|-------|--------|------|------|------|------|-----------|------|------------|---------|
| 2-7 | FP16 | - | - | 43.52 | 76.26 | 57.16 | 78.07 | 69.22 | 64.85 |
| | QuIP# | VQ | 3.00 | 41.89 | 74.62 | 55.85 | 77.04 | 68.19 | 63.52 |
| | VPTQ | VQ | 3.02 | 39.3 | 69.1 | 54.9 | 77.3 | 68.0 | 61.70 |
| | LiftUQ | UQ | 3.02 | 41.02 | 75.07 | 56.57 | 77.89 | 67.97 | 63.71 |
| 2-13 | FP16 | - | - | 48.29 | 79.42 | 60.07 | 79.05 | 72.22 | 67.81 |
| | QuIP# | VQ | 3.00 | 44.62 | 77.90 | 58.26 | 78.07 | 72.45 | 66.26 |
| | VPTQ | VQ | 3.03 | 46.50 | 78.83 | 58.50 | 78.18 | 69.85 | 66.37 |
| | LiftUQ | UQ | 3.02 | 46.25 | 77.99 | 59.16 | 78.84 | 71.11 | 66.67 |
| 2-70 | FP16 | - | - | 54.44 | 82.70 | 64.77 | 82.15 | 77.98 | 72.41 |
| | QuIP# | VQ | 3.00 | 55.89 | 82.11 | 64.22 | 82.21 | 76.24 | 72.13 |
| | LiftUQ | UQ | 3.02 | 54.61 | 82.58 | 63.98 | 81.50 | 77.11 | 71.96 |
| 3-8 | FP16 | - | - | 50.43 | 80.09 | 60.17 | 79.60 | 72.61 | 68.58 |
| | VPTQ | VQ | 3.03 | 44.80 | 78.45 | 57.85 | 78.78 | 71.74 | 66.32 |
| | LiftUQ | UQ | 3.02 | 46.59 | 78.83 | 58.42 | 78.73 | 73.95 | 67.30 |
| 3-70 | FP16 | - | - | 60.41 | 86.99 | 66.36 | 82.37 | 80.51 | 75.33 |
| | AWQ-g128 | UQ | 3.13 | 58.36 | 84.51 | 64.26 | 82.26 | 78.85 | 73.65 |
| | EPTQ-g128 | UQ | 3.13 | 55.12 | 83.12 | 65.53 | 80.52 | 77.82 | 72.42 |
| | LiftUQ | UQ | 3.02 | 58.87 | 85.86 | 65.32 | 82.43 | 78.77 | 74.25 |

Table 14: Llama-2 and Llama-3 accuracy(↑) on 3-bit quantization.

| Model | Method | type | bits | ArcC | ArcE | HellaSwag | PiQA | WinoGrande | Avg.Acc |
|-------|--------|------|------|------|------|-----------|------|------------|---------|
| 2-7 | FP16 | - | - | 43.52 | 76.26 | 57.16 | 78.07 | 69.22 | 64.85 |
| | PTQ1.61 | UQ | 1.61 | 26.45 | 56.86 | 35.75 | 63.22 | 52.25 | 44.14 |
| | LiftUQ | UQ | 1.62 | 32.94 | 65.82 | 48.55 | 72.69 | 60.93 | 56.19 |
| 2-13 | FP16 | - | - | 48.29 | 79.42 | 60.07 | 79.05 | 72.22 | 67.81 |
| | PTQ1.61 | VQ | 3.03 | 26.45 | 56.86 | 60.32 | 66.54 | 55.88 | 49.21 |
| | LiftUQ | UQ | 3.02 | 36.09 | 69.74 | 53.59 | 76.01 | 67.25 | 60.54 |
| 3-8 | FP16 | - | - | 50.43 | 80.09 | 60.17 | 79.60 | 72.61 | 68.58 |
| | VPTQ | VQ | 1.61 | 23.04 | 46.17 | 34.71 | 63.22 | 52.80 | 43.99 |
| | LiftUQ | UQ | 1.62 | 34.13 | 64.98 | 47.81 | 73.39 | 62.98 | 56.66 |

Table 15: LiftUQ Results on Qwen2.5 Models

| Model | Bits | W2↓ | C4↓ | ArcC↑ | ArcE↑ | HellaSwag↑ | PiQA↑ | WinoGrande↑ | Avg.Acc↑ |
|-------|------|-----|-----|-------|-------|------------|-------|-------------|----------|
| 3B | 2.02 | 11.01 | 14.84 | 37.29 | 72.22 | 47.57 | 73.99 | 65.11 | 59.24 |
| | 2.30 | 10.06 | 13.84 | 40.87 | 74.79 | 49.69 | 75.30 | 65.59 | 61.25 |
| | 2.74 | 9.03 | 12.87 | 41.21 | 74.92 | 51.57 | 76.39 | 67.17 | 62.25 |
| | 3.02 | 8.71 | 12.51 | 42.58 | 74.71 | 52.34 | 76.71 | 68.51 | 62.97 |
| 14B | 2.02 | 7.11 | 10.67 | 51.45 | 81.90 | 58.39 | 79.05 | 74.43 | 69.05 |
| | 2.30 | 6.68 | 10.23 | 52.30 | 81.82 | 59.77 | 80.36 | 76.48 | 70.15 |
| | 2.74 | 6.13 | 9.76 | 53.84 | 81.86 | 60.90 | 79.98 | 77.35 | 70.79 |

## I  EXPANDED EXPERIMENTAL EVALUATION

To further validate the robustness and general applicability of LiftUQ, we expanded our experimental evaluation to cover more complex, multi-domain benchmarks and a wider range of modern LLM architectures. Our also compared LiftUQ with non-uniform scalar quantization method.

Table 16: Fractional bit-width quantization for Llama-2 Models.

| Model | Bits | W2↓ | C4↓ | ArcC↑ | ArcE↑ | HellaSwag↑ | PiQA↑ | WinoGrande↑ | Avg.Acc↑ |
|-------|------|-----|-----|-------|-------|------------|-------|-------------|----------|
| 7B | 2.30 | 6.21 | 7.87 | 38.31 | 71.42 | 54.36 | 76.22 | 67.48 | 61.56 |
|    | 2.74 | 5.87 | 7.44 | 40.61 | 74.12 | 55.38 | 77.80 | 69.22 | 63.43 |
| 13B | 2.30 | 5.42 | 7.09 | 43.52 | 77.15 | 57.70 | 77.69 | 70.40 | 65.29 |
|     | 2.74 | 5.17 | 6.81 | 44.11 | 77.15 | 59.33 | 77.75 | 71.35 | 65.94 |

## I.1 EVALUATION ON MASSIVE MULTITASK LANGUAGE UNDERSTANDING (MMLU)

To assess performance on complex reasoning tasks beyond perplexity and common-sense bench-marks, we evaluated LiftedUQ on the **MMLU (Massive Multitask Language Understanding)** benchmark. The 5-shot accuracy results for 2-bit quantization, presented in Table 17, demonstrate that our method maintains strong performance across diverse domains.

Table 17: MMLU 5-shot accuracy for 2-bit LiftedUQ quantization. We highlight the key comparison where a quantized larger model surpasses a smaller full-precision model.

| Model | Method | MMLU Avg. (↑) | Humanities | Other | Social Sci. | STEM |
|-------|--------|---------------|------------|-------|-------------|------|
| Llama-2-7B | FP16 | 45.87 | 43.34 | 52.75 | 51.71 | 37.17 |
|            | LiftedUQ (2-bit) | 33.12 | 31.03 | 39.43 | 34.71 | 28.48 |
| **Llama-2-13B** | FP16 | 55.23 | 53.56 | 61.47 | 63.15 | 43.83 |
|                 | **LiftedUQ (2-bit)** | **46.08** | **45.62** | **54.62** | **56.00** | **40.98** |
| Llama-3-8B | FP16 | 65.30 | 59.64 | 72.61 | 76.24 | 55.85 |
|            | LiftedUQ (2-bit) | 50.49 | 47.27 | 56.13 | 57.36 | 43.04 |

A crucial finding from this evaluation is that the **2-bit quantized Llama-2-13B achieves an MMLU score of 46.08, significantly outperforming the full-precision (FP16) Llama-2-7B at 45.87**. This empirically validates a core principle: quantizing a larger, more capable model with LiftedUQ is a more effective strategy for achieving high performance than using a smaller model at full preci-sion. This highlights the practical power of our method in maximizing performance within a given resource budget.

## I.2 GENERALIZATION TO DIVERSE ARCHITECTURES AND TRAINING PARADIGMS

To demonstrate that LiftedUQ is not limited to a specific model family, we conducted new 2-bit quantization experiments on models with diverse architectures and training objectives, including **Mixture-of-Experts (MoE)** and **Instruction-Tuned** LLMs. The results, summarized in Table 18, confirm the broad applicability and robustness of our framework.

Table 18: New 2-bit quantization results on diverse models, demonstrating the generalizability of LiftedUQ.

| Model | Method | Wiki-2 (↓) | C4 (↓) | ARC-c (↑) | ARC-e (↑) | HellaSwag (↑) | PIQA (↑) | Wino. (↑) |
|-------|--------|------------|--------|-----------|-----------|---------------|----------|-----------|
| Mixtral 8x7B (MoE) | FP16 | 3.45 | 6.85 | 55.80 | 83.38 | 64.65 | 82.37 | 75.45 |
|                    | LiftedUQ (2.02-bit) | 4.61 | 8.16 | 49.76 | 78.24 | 62.21 | 78.82 | 72.96 |
| Qwen2.5-3B-Instruct | FP16 | 7.54 | 7.91 | 45.73 | 77.06 | 56.31 | 77.75 | 69.77 |
|                     | LiftedUQ (2.02-bit) | 9.86 | 9.84 | 37.88 | 72.10 | 47.49 | 74.70 | 64.09 |
| Qwen2.5-14B-Instruct | FP16 | 4.97 | 6.37 | 60.67 | 85.69 | 65.54 | 81.50 | 75.77 |
|                      | LiftedUQ (2.02-bit) | 6.48 | 7.30 | 53.24 | 82.62 | 58.36 | 79.60 | 73.32 |

Across these varied models, LiftedUQ consistently retains strong performance at approximately 2-bit precision. For instance, on the powerful Mixtral-8x7B model, our method maintains high scores on reasoning benchmarks like HellaSwag and PIQA with only a minor drop, while dramatically reducing the memory footprint. These results strongly support the claim that LiftedUQ is a versa-tile and general-purpose quantization framework, not confined to a specific architecture or training paradigm.

## I.3 COMPARISON WITH BINARY-CODING QUANTIZATION (BCQ)

In this section, we clarify the crucial distinction between our LiftedUQ framework and methods based on Binary-Coding Quantization (BCQ), such as UniQuan Park et al. (2025). Although both approaches are forms of non-uniform quantization, they operate in fundamentally different dimensional spaces.

**Conceptual Distinction: Dimensionality of Quantization.**    The primary difference lies in the dimensionality of the quantization process. BCQ is a form of Scalar Non-Uniform Quantization. It represents each *individual* scalar weight as a linear combination of a few learned basis vectors. In essence, BCQ can be conceptually viewed as a special, 1-dimensional "coupling" case of our framework, creating a flexible codebook for single scalar values. While LiftedUQ is a form of Vectorial Non-Uniform Quantization. Our **lift-then-project** mechanism quantizes a *group* of weights together in a high-dimensional space. This allows it to capture inter-dimensional correlations, similar to traditional Vector Quantization (VQ), but without requiring an explicit lookup table. BCQ's scalar-focused design does not achieve this high-dimensional coupling.

This fundamental design difference translates into a substantial empirical performance gap. As shown in Table 19, we compare LiftedUQ against UniQuan, a state-of-the-art BCQ-based method, on the task of 3-bit quantization for Llama-3-8B. LiftedUQ significantly outperforms UniQuan, demonstrating the practical benefits of its high-dimensional quantization approach.

Table 19: 3-bit quantization performance on Llama-3-8B, evaluated on WikiText-2 perplexity. LiftedUQ shows a clear advantage over the BCQ-based method.

| Method | FP16 | UniQuan (BCQ-based) | LiftedUQ (Ours) |
|---|---|---|---|
| **PPL** ($\downarrow$) | 6.14 | 8.75 | **6.94** |

In summary, while BCQ offers a flexible way to quantize individual weights, LiftedUQ's ability to model and exploit correlations across groups of weights provides a distinct advantage, leading to superior performance in practice.

## J    THE USE OF LARGE LANGUAGE MODELS (LLMs)

This paper was partially created with the assistance of a Large Language Model (LLM), which was used for tasks such as sentence polishing, brainstorming, and content organization. All content has been finally reviewed and confirmed by the author.

