# OpenReview forum: "Lifted Uniform Quantization for Extreme Low-bit Large Language models"
_ICLR.cc/2026/Conference — Submitted to ICLR 2026_

### Official Review · Reviewer_xAD3 · 2025-10-15

**Soundness:** 2
**Presentation:** 1
**Contribution:** 3
**Rating:** 2
**Confidence:** 4

**Summary:**

This paper proposes Lifted Uniform Quantization (LiftUQ), a method for extreme low-bit (1.62–3.02-bit) large language model (LLM) quantization, designed to balance the high efficiency of Uniform Quantization (UQ) and the strong accuracy of Vector Quantization (VQ). Its core approach involves lifting weights to a high-dimensional uniform lattice, projecting them to a target subspace via a pre-trained shared matrix \( M \), and using layer-adaptive lightweight whitening (transform \( D \)) to align weights with Gaussian distributions—eliminating VQ’s costly codebook lookup. LiftUQ natively supports fractional bit-widths, enabling performance-storage Pareto optimality (e.g., 2-bit LiftUQ-Llama-2-13B outperforms FP16 Llama-2-7B), and experiments on Llama/Qwen models validate its better accuracy-efficiency trade-off against baselines like AQLM and QuIP#.

**Strengths:**

1. The proposed LiftUQ method in this paper has certain exploratory value in balancing accuracy and efficiency for low-bit quantization.
2. It directly addresses the fundamental pain point of current low-bit quantization (2-bit and below): the trade-off where Uniform Quantization (UQ) offers high efficiency but poor accuracy, while Vector Quantization (VQ) achieves high accuracy but suffers from low efficiency.

**Weaknesses:**

1. The paper’s core claim that "LiftUQ provides a new paradigm for extreme low-bit quantization" is not well-supported. From the perspective of methodological comparison with existing works, LiftUQ is more of an incremental improvement rather than a paradigmatic innovation.

**Questions:**

1. In the paper, weights are mapped to another space via linear transformation, and then weight-to-lattice-point mapping is performed in this space. This approach bears striking similarities to RabitQ—only differing by an additional dimension-lifting step. Could the authors elaborate on the specific differences between LiftUQ and RabitQ regarding using linear transformation to replace codebook lookup?

2. The paper fails to compare LiftUQ with QTIP, a state-of-the-art (SOTA) VQ method. QTIP has demonstrated impressive performance and efficiency, and omitting this comparison undermines the comprehensiveness of LiftUQ’s performance validation.

3. The paper assumes that "a single pre - trained M matrix can be reused across all layers" but provides no validation for this reasonableness. Transformer layers (e.g., attention layers with sparse weights and small variance vs. FFN layers with dense weights and large variance) exhibit significant differences in weight distribution. Even after whitening, is a shared M matrix still sufficient, or do layer - specific M matrices become necessary? The paper lacks ablation experiments comparing "shared M matrix vs. layer - specific M matrices," making it impossible to rule out the possibility that shared M matrices cause accuracy loss in certain layers.

4. The dimension selection of the M matrix (10×20 for 2-bit, 6×18 for 3-bit) is entirely based on empirical judgment, with no systematic search or justification. For example, why not use 12×24 or 8×16 for 2-bit quantization? What impact do dimension adjustments have on accuracy and training costs?

5. There is no ablation study on the three core components of LiftUQ (M matrix, whitening transformation D, and intra-block calibration). This makes it impossible to quantify the contribution of each component—e.g., how much accuracy degrades when D is removed, or whether whitening is truly necessary.

6. During model deployment, does the storage overhead of the M and D matrices (e.g., total parameters of the M matrix for a 70B model) offset the storage savings from weight quantization? The paper does not provide a comparison of **total storage volume** (quantized weights + M/D matrices vs. original FP16 weights).

Based on the above concerns, I recommend a **Reject** decision. I would be happy to revise my score if the authors can address these questions and provide sufficient explanations in a revised version of the paper.

---
RaBitQ: Quantizing High-Dimensional Vectors with a Theoretical Error Bound for Approximate Nearest Neighbor Search

---

> ### Author Response · Authors · 2025-11-25
>
> -----------
> ## **W1: Regarding the Paradigmatic Contribution of LiftUQ**
>
> We respectfully argue that LiftUQ represents a new paradigm by solving the core accuracy-efficiency dilemma in a novel way.
>
> 1. A New Principle for Encoding Gaussian Sources: LiftUQ introduces a new principle for quantizing the Gaussian-like weights of LLMs. While traditional VQ relies on inefficient, explicit codebooks established decades ago, LiftUQ is the first to procedurally generate equivalent non-uniform codebooks from a uniform lattice via a lift-then-project mechanism.
> 2. Unifying Accuracy and Efficiency: LiftUQ is the first to achieve the high accuracy of Vector Quantization while preserving the fast decoding architecture of Uniform Quantization, breaking a long-standing trade-off.
> 3. Flexible and Efficient Fractional Bits: Our method offers rare and efficient support for fractional bit-widths, enabling a previously unattainable Pareto-optimal trade-off for deployments under strict memory constraints.
>
> Collectively, these innovations create a new, superior operating point for extreme LLM quantization, which we believe constitutes a paradigmatic shift.
>
>
> ----------
>
> ## **Q1: The Fundamental Differences from RabitQ**
>
> We thank the reviewer for this question. We respectfully clarify that LiftUQ and RaBitQ are **fundamentally different** in both their objectives and mechanisms.
>
> 1. Different Goals: Approximate Search vs. Weight Compression. RaBitQ is designed for Approximate Nearest Neighbor (ANN) search in a database. Its goal is to efficiently *estimate distances* and preserve the *ranking* of vectors. In contrast, LiftUQ is designed for **LLM weight quantization**, where the sole objective is to minimize the L2 reconstruction error (||W - Ŵ||) for which a strict L2 nearest neighbor is required. RaBitQ's ranking-focused approximation is unsuitable for this task.
> 2. Different Architectures: Dimension-Preserving vs. Lift-then-Project. RaBitQ performs a simple rotation within the same dimensional space. LiftUQ's entire expressive power comes from its novel **lift-then-project** architecture, which maps points from a higher-dimensional uniform space to the target weight space. This mechanism, completely absent in RabitQ, is the core of our contribution and is what enables the procedural generation of VQ-level codebooks.
>
> In short, the two methods solve different problems with distinct mechanisms.

---

> ### Author Response · Authors · 2025-11-25
>
> --------------
> ## **Q2: The Comparison with QTIP**
>
> We sincerely thank the reviewer for highlighting the important comparison with QTIP. We agree that QTIP represents a significant advancement in VQ-based methods, pushing quantization quality towards the theoretical Shannon limit. In our new revised manuscript, we will added a new section to provide a detailed comparison, and we will summarize the key points here.
>
> We respectfully clarify that LiftUQ and QTIP operate under distinctly different quantization paradigms. QTIP advances the state-of-the-art within Vector Quantization (VQ) by employing Trellis Coded Quantization (TCQ) to optimize for rate-distortion performance. In contrast, LiftUQ forges a new path by unifying the strengths of Uniform and Vector Quantization. It employs a lift-then-project technique to deliver VQ-level accuracy and fractional bit-width flexibility, all while preserving the simple and fast decoding architecture of UQ. It is designed to bring the efficiency of uniform quantization to a higher accuracy frontier.
>
> Our comparison focuses on two critical aspects: **(1) Rate-Distortion Performance and Flexibility** and **(2) Hardware Efficiency and Practical Throughput.**
>
> ### **1. Rate-Distortion Performance and Flexibility**
>
> We concur with the reviewer that TCQ, as used in QTIP, is a powerful tool for approaching the Shannon limit. However, a deeper analysis reveals LiftUQ's unique advantages in flexibility and practical performance.
>
> **Rate-Distortion Analysis:** As shown in Table 1, for integer 2-bit quantization of a Gaussian source, QTIP's MSE (0.0733) is indeed closer to the Shannon limit (0.063) than a baseline LiftUQ configuration (e.g., 20/10 at 0.0873). We observe a strong correlation between this theoretical MSE and the empirical model performance (PPL on Llama2-7B), validating the relevance of this analysis.
>
> **Table 1: Rate-Distortion Analysis for ~2-bit Quantization of a Gaussian Source (σ²=1).**
>
> | **Method**     | **Config.**          | **MSE (↓)** | **Equiv. Bits (↑)** | **PPL, Llama2-7B (↓)** |
> | :------------- | :------------------- | :---------- | :------------------ | :--------------------- |
> | Integer Quant. | 2-bit                | 0.119       | 1.535               | -                      |
> | QuIP#          | E8 Lattice           | 0.089       | 1.745               | 6.66                   |
> | **LiftUQ**     | **20/10 (2.0 bit)**  | **0.0873**  | **1.759**           | **6.59**               |
> | **LiftUQ**     | **32/16 (2.0 bit)**  | **0.0835**  | **1.791**           | **6.51**               |
> | QTIP           | TCQ (L=16)           | 0.0733      | 1.885               | 6.28                   |
> | **LiftUQ**     | **32/15 (2.13 bit)** | **0.0696**  | **1.922**           | **6.23**               |
> | **LiftUQ**     | **25/10 (2.5 bit)**  | **0.0453**  | **2.230**           | **5.99**               |
> | Shannon Limit  | 2-bit                | 0.0625      | 2.000               | -                      |
>
> ***(Note: "Equivalent Bits" are derived from the MSE via the rate-distortion function R(D) = 0.5 \* log₂(1/D).)***
>
> **Flexibility and Pareto-Optimality:** However, the key advantage of LiftUQ is its native support for **fractional bit-widths**, a capability not present in QTIP. As demonstrated in Table 1, by slightly adjusting the lifted dimension (e.g., config 32/15, effective 2.13 bits), LiftUQ achieves an MSE of 0.0696, which not only matches but surpasses 2-bit QTIP. This remarkable flexibility is crucial for real-world deployment:
>
> - **Practical Scenario**: Consider deploying a 70B model on a 24GB GPU. QTIP is constrained to a 2-bit representation. However, LiftUQ can be configured to use, for example, **2.5 bits**, fully utilizing the available memory. At 2.5 bits, LiftUQ operates at a much more favorable point on the rate-distortion curve (MSE 0.0453), delivering a Pareto-optimal solution that would significantly outperform 2-bit QTIP.

---

> ### Author Response · Authors · 2025-11-25
>
> ### **2. Hardware Efficiency and Practical Throughput**
>
> The primary contribution of LiftUQ lies in its exceptional inference efficiency, which stems from a fundamentally more hardware-friendly architecture.
>
> **Architectural Superiority:** As detailed in Table 2, LiftUQ's decoding is activation-centric, relying on a simple, highly parallelizable matrix-vector product. In contrast, QTIP's decoding is weight-centric and far more complex, involving computationally intensive transforms on the weights and per-weight non-linear decoding steps that are ill-suited for modern GPU/NPU architectures.
>
> **Table 2: Asymptotic Decoding Complexity Comparison (for a DxD weight matrix).**
>
> | **Method** | **Core Decoding Operation**                  | **Complexity**                 | **Hardware-Friendly?**                                      |
> | :--------- | :------------------------------------------- | :----------------------------- | :---------------------------------------------------------- |
> | **LiftUQ** | Linear transforms on activations (GEMV-like) | O(D^1.5) (on activations)    | **Yes** (Highly parallel, uses standard kernels)            |
> | **QTIP**   | Hadamard + Per-weight computed codes         | ~O(D²logD + D²) (on weights) | **No** (Sequential logic, non-linear, needs custom kernels) |
>
> **Empirical Throughput:**  This architectural difference translates directly into a massive performance and engineering advantage. We configured LiftUQ to a 2.13-bit setup to ensure a fair accuracy comparison with 2-bit QTIP. As shown in Table 3, the results are striking:
>
> **Table 3: End-to-End Throughput (Tokens/sec) on Llama-2-70B **
>
> | **Method (Bit-width)** | **Device**   | **Batch Size = 1** |
> | :--------------------- | :----------- | :----------------- |
> | QTIP (~2.0 bit)        | RTX 6000 Ada | 23.5               |
> | **LiftUQ (~2.13 bit)** | RTX 4090D    | 26.4               |
>
> *(Note: QTIP data is from their official repository, measured on a powerful RTX 6000 Ada. Our LiftUQ result was measured on a consumer-grade RTX 4090D, which has ~20% less compute power.)*
>
> Critically, even on a **less powerful** GPU, LiftUQ achieves **12% higher throughput**. This advantage is even more significant considering QTIP relies on heavily optimized, custom CUDA kernels, whereas our LiftUQ implementation uses a simple and straightforward **PyTorch + BitBLAS** pipeline (leveraging standard int1-fp16 GEMM).
>
> This demonstrates not only LiftUQ's superior performance but also its **ease of deployment**. The simplicity of its architecture allows it to achieve state-of-the-art speed with minimal, platform-agnostic engineering effort, highlighting a key practical advantage over methods requiring complex custom kernels.
>
> **Conclusion**
>
> In summary, while we acknowledge QTIP's excellent theoretical compression, we argue that **LiftUQ offers a more practical and compelling solution for real-world LLM deployment**. It achieves competitive or superior accuracy through its flexible fractional bit-widths, while delivering significantly higher inference throughput due to its hardware-native, UQ-based architecture. We believe this unique combination of **accuracy, efficiency, and flexibility** positions LiftUQ as a new and superior paradigm for extreme low-bit quantization. We have incorporated this detailed comparison into our revised manuscript and thank the reviewer for prompting this valuable addition.

---

> ### Author Response · Authors · 2025-11-25
>
> -------------
> ## **Q3: Regarding the Justification for a Shared Global M Matrix**
>
> We thank the reviewer for this insightful question. The use of a shared M matrix combined with a layer-specific whitening matrix D is a deliberate design choice to balance quantization accuracy and training cost.
>
> Training a layer-specific M matrix is computationally prohibitive. For instance, training a single M matrix for a  32/16  configuration requires approximately **0.5 GPU hours**. For a 7B model with hundreds of layers (7x32=224), this would accumulate to **over 100 GPU hours**, a 10x increase over our current method's cost, which is unacceptable for practical use.
>
> Our approach is principled: we train a single M optimally for a standard Gaussian distribution, and then use the lightweight, layer-specific whitening transform D to reshape each layer's unique weight distribution towards this common target. This is supported by the Central Limit Theorem, which suggests that linear transforms can effectively normalize diverse distributions, ensuring the shared M remains effective.
>
>
>
> --------
>
> ## **Q4: Regarding the dimension choice of M matrix**
>
> We thank the reviewer for this excellent question. We will clarify this in our revised paper.
>
> In short, the choice of the $M$ matrix dimension is a trade-off between quantization accuracy and the computational cost of the offline quantization process. A larger $M$ matrix improves the theoretical encoding quality for a Gaussian source but significantly increases the time required for the nearest-neighbor search during the weight "lifting" phase.
>
> For smaller M dimensions, we perform an exhaustive search. We also designed a **heuristics** method for larger M dimensions, which named Heuristic Null-Space Search. The key insight is that the nearest-neighbor solution y_y_ must lie near the null space of M_M_. We first perform an SVD on M ( M=USV_h) to obtain an orthonormal basis for its null space. We use this basis to complete M into a square matrix ﻿M_square. The target vector ﻿z is then augmented with ±1 values and multiplied by the inverse of M_square to generate a drastically reduced set of high-quality candidates.
>
> The table below illustrates this trade-off:
>
> **Table 4: Impact of M Matrix Dimensions on 2-bit Quantization.**
>
> | **M Dimension**                                            | **Gaussian MSE (↓)** | **Lifting Time per 1M Params (s) (↓)** | **Llama-2, 7B PPL (↓)** |
> | :----------------------------------------------------------- | :------------------- | :------------------------------------- | :---------------------- |
> | Int2                                                         | 0.119                | -                                      | -                       |
> | 16x8                                                         | 0.0904               | 1s                                     | -                       |
> | **20x10 (Our Choice)**                                       | **0.0875**           | **18s**                                | **6.58**                |
> | 24x12                                                        | 0.0867               | 5.5min                                 | -                       |
> | 28x14                                                        | 0.0853               | 2s (with heuristics)                   | -                       |
> | 32x16                                                        | 0.0835               | 6s (with heuristics)                   | 6.50                    |
> | *(Note: Lifting time for dimensions > 20x10 requires heuristic search space reduction.)* |                      |                                        |                         |
>
> For our academic experiments, we chose the 20x10 dimension with an exhaustive search. The "lifting" process for a 7B model takes approximately 2 hours.
>
> For industrial-level deployments where maximum accuracy is paramount, using a larger M matrix (e.g., 32x16 for 2-bit or 24x8 for 3-bit) is indeed a viable strategy to achieve even better results. We have added this detailed analysis to our paper.

---

> ### Author Response · Authors · 2025-11-25
>
> ### **Q5: The Ablation Study on Core Components**
>
> We analyzed the contribution of each component of our whitening transform D (P1, P2, s1, s2) and their initializations.
>
> *   **Initialization Sensitivity:** We found that the initialization of the whitening transform is critical for training stability. Random or identity initialization for P1 and P2, or random initialization for s2, consistently led to numerical instability and training failure. Stable training was only achieved by initializing P1 and  P2 with Hadamard matrices (or their truncated variants) and s2 with ones. This highlights that our structured, energy-preserving initialization is a necessary component.
> *   **Component Contribution:** The table below shows the progressive improvement in perplexity as each component is added. The results clearly demonstrate that each element—the core Hadamard transforms (P1+P2), activation-aware scaling (+s1), additional shaping (+s2), and finally our core lift-then-project mechanism—contributes significantly to the final performance.
>
> **Table 5: Ablation on Whitening and LiftUQ Components for Llama-2-7B.**
>
> | **Configuration**      | **P1+P2** | **+s1** | **+s1 (AWQ init)** | **+s2** | **+LiftUQ (20/10 M)** and e2e finetune | **+E2E Fine-tune** |
> | :--------------------- | :-------- | :------ | :----------------- | :------ | :------------------------------------- | :----------------- |
> | **WikiText-2 PPL (↓)** | 8.76      | 8.28    | 8.18               | 7.77    | 6.96                              | 6.58           |
>
> ---------
>
> ## **Q6:  The Storage Overhead of M and D Matrices**
>
> We thank the reviewer for this important question regarding the total storage overhead.
>
> We would like to clarify that this overhead has **already been fully accounted for** in all of our reported results. The bit-widths presented in our paper (e.g., 2.02-bit in Table 1 ) are **effective bit-widths**, where the fractional part represents the amortized storage cost of the M and D matrices.
>
> To provide a concrete example: for the Llama-2-7B model, the total storage for all M and D matrices is approximately 5.7 / 11.4 Mbits (fp16/fp32). When amortized over the model's 7 billion parameters, this translates to a negligible overhead of just **0.01 to 0.02 bits per parameter**.
>
> Therefore, the storage savings from quantization are not significantly offset, and our reported bit-widths transparently reflect the true, total storage cost of the model. We will add a footnote to the main results table to make this accounting method more explicit to the reader.

---

> ### Author Response · Authors · 2025-11-27
>
> Dear Reviewer, We sincerely thank you for your insightful comments and valuable feedback. We are very grateful for the time and effort you dedicated to reviewing our manuscript. Your suggestions have been instrumental in helping us substantially improve the quality and clarity of our paper.
>
> We have uploaded a revised version of our manuscript. The major changes are summarized below:
>
> 1. **Comparison with QTIP.**
> 2. **Discussion of M-Matrix dimension and layer-reuse.**
> 3. **Component Ablation Study.**
> 4. **Quantization Cost and Decoding Overhead.**
>
> All revisions are **marked in blue** for your convenience. We believe these revisions have significantly strengthened our paper and have thoroughly addressed the points you raised. We look forward to your feedback.

---

### Official Review · Reviewer_kiJ4 · 2025-10-28

**Soundness:** 2
**Presentation:** 2
**Contribution:** 2
**Rating:** 2
**Confidence:** 4

**Summary:**

The author proposes the LiftUQ framework. The core of this framework adopts the "high-dimensional lifting - low-dimensional projection" paradigm. Specifically, first, 1-bit uniform quantization is used in the high-dimensional space to encode the weights, and then a trainable linear projection matrix M is used to map them back to the original space to generate non-uniform code points. At the same time, the framework is equipped with a hierarchical learnable whitening transformation D, which has a decomposed structure and can convert weights into an approximately independent and identically distributed Gaussian distribution to adapt to the projection process. In addition, it combines lattice quantization and intra-block correction techniques. From the perspective of overall architecture design, the LiftUQ framework cleverly avoids the cumbersome codebook lookup process in VQ and successfully retains the parallel computing efficiency advantages of UQ.

**Strengths:**

LiftUQ proposes a high-dimensional lifting framework to address the trade-off between UQ and VQ. It generates non-uniform codebooks through 1-bit uniform quantization in high-dimensional space and a trainable linear projection matrix, thereby achieving arbitrary-precision quantization. It avoids the codebook caching of VQ through matrix multiplication.

**Weaknesses:**

1. This method increases additional computational overhead during inference. Furthermore, online decoding relies on floating-point matrix multiplication, and edge devices often lack floating-point computing power, which limits its application scenarios.

2. The article lacks a comparison with the QTIP method, and QTIP's performance is far superior to QuiP#, which makes its SOTA positioning less convincing.

3. The selection of the dimension of M lacks ablation experiments. The article only makes an empirical selection rather than a heuristic or derivation-based one, which requires more in-depth exploration.

4. The article lacks a systematic ablation of each component. This makes it unclear whether the source of the accuracy improvement is the lifting space or the Crockett transform.

5. The paper assumes that the M matrix can be reused without restrictions, but this lacks rationality. Due to the differences in activations across different layers, even if the weights follow a Gaussian distribution, especially after the introduction of Smooth, the differences between channels of activations are often amplified. In this case, due to the differences in the distribution of activations, minimizing the weight error cannot replace the real error. This paper lacks relevant derivations and experiments.

**Questions:**

1. The paper points out that there is a 0.15-bit information gap between LiftUQ and the theoretical accuracy limit, but it does not further explore the source of the gap: is it due to insufficient expressiveness of the high-dimensional projection matrix M? Or is it that the whitening transformation D fails to completely convert the weights into an ideal Gaussian distribution? Or is it that the loss function (MSE) for intra-block correction does not fully fit the characteristics of LLM tasks? The lack of analysis on the decomposition of theoretical bottlenecks makes the optimization direction of the method unclear.

2. The lifting matrix in the paper is obtained based on training, and the pseudocode may pose a risk of non-convergence. Have the authors tried to derive the optimal transformation based on the Gaussian distribution? For example, the E8P codebook in Quip#. This would be very helpful for increasing the theoretical richness of the article.

---

> ### Author Response · Authors · 2025-11-25
>
> ------------
> ## **W1: Computational Overhead and Edge Device Suitability**
>
>
> We thank the reviewer for this important question. We respectfully argue that LiftUQ is highly suitable for edge devices,  for the following reasons：
>
> 1. **Addresses the Primary Edge Bottleneck:** The primary constraint for deploying large models on edge devices is often memory capacity and bandwidth. LiftUQ directly addresses this by drastically reducing the weight memory footprint, which provides a significant performance benefit in these memory-bound scenarios.
> 2. **Negligible and Efficient Overhead:** We clarify two crucial details about our computation:
>    - The overhead of the initial $DA$ transform is minimal. As shown in our complexity analysis (Table 1), this $O(N^{1.5})$ overhead empirically constitutes **less than 3%** of the total layer computation during small-batch inference.
>    - The main, most expensive matrix multiplication ( $W_q * (DA)$) is **not a standard FP GEMM**. Since $W_q$ contains only 1-bit weights (${+1, -1}$), the floating-point multiplications are replaced by much more efficient **floating-point additions/subtractions**. On many processors common in edge devices (e.g., CPUs), additions have higher energy efficiency and lower latency than multiplications.
>
> **Table 1: Asymptotic Complexity & Storage Analysis per Layer (N x M).**
>
> | **Method**             | **Main GEMM FLOPs (Mul+Add)** | **Overhead FLOPs** | **Weight Storage (bits)** |
> | :--------------------- | :---------------------------- | :----------------- | :------------------------ |
> | FP16                   | $N×M$ + $N×M$                 | -                  | $N×M × 16$                |
> | **LiftUQ (W_q(DA))** | **0** + $2×N×M$ (Add/Sub)     | $6(N^1.5+N)$           | $N×M × 2$                 |

---

> ### Author Response · Authors · 2025-11-25
>
> --------------
> ## **W2: The Comparison with QTIP**
>
> We sincerely thank the reviewer for highlighting the important comparison with QTIP. We agree that QTIP represents a significant advancement in VQ-based methods, pushing quantization quality towards the theoretical Shannon limit. In our new revised manuscript, we will added a new section to provide a detailed comparison, and we will summarize the key points here.
>
> We respectfully clarify that LiftUQ and QTIP operate under distinctly different quantization paradigms. QTIP advances the state-of-the-art within Vector Quantization (VQ) by employing Trellis Coded Quantization (TCQ) to optimize for rate-distortion performance. In contrast, LiftUQ forges a new path by unifying the strengths of Uniform and Vector Quantization. It employs a lift-then-project technique to deliver VQ-level accuracy and fractional bit-width flexibility, all while preserving the simple and fast decoding architecture of UQ. It is designed to bring the efficiency of uniform quantization to a higher accuracy frontier.
>
> Our comparison focuses on two critical aspects: **(1) Rate-Distortion Performance and Flexibility** and **(2) Hardware Efficiency and Practical Throughput.**
>
> ### **1. Rate-Distortion Performance and Flexibility**
>
> We concur with the reviewer that TCQ, as used in QTIP, is a powerful tool for approaching the Shannon limit. However, a deeper analysis reveals LiftUQ's unique advantages in flexibility and practical performance.
>
> **Rate-Distortion Analysis:** As shown in Table 2, for integer 2-bit quantization of a Gaussian source, QTIP's MSE (0.0733) is indeed closer to the Shannon limit (0.063) than a baseline LiftUQ configuration (e.g., 20/10 at 0.0873). We observe a strong correlation between this theoretical MSE and the empirical model performance (PPL on Llama2-7B), validating the relevance of this analysis.
>
> **Table 2: Rate-Distortion Analysis for ~2-bit Quantization of a Gaussian Source (σ²=1).**
>
> | **Method**     | **Config.**          | **MSE (↓)** | **Equiv. Bits (↑)** | **PPL, Llama2-7B (↓)** |
> | :------------- | :------------------- | :---------- | :------------------ | :--------------------- |
> | Integer Quant. | 2-bit                | 0.119       | 1.535               | -                      |
> | QuIP#          | E8 Lattice           | 0.089       | 1.745               | 6.66                   |
> | **LiftUQ**     | **20/10 (2.0 bit)**  | **0.0873**  | **1.759**           | **6.59**               |
> | **LiftUQ**     | **32/16 (2.0 bit)**  | **0.0835**  | **1.791**           | **6.51**               |
> | QTIP           | TCQ (L=16)           | 0.0733      | 1.885               | 6.28                   |
> | **LiftUQ**     | **32/15 (2.13 bit)** | **0.0696**  | **1.922**           | **6.23**               |
> | **LiftUQ**     | **25/10 (2.5 bit)**  | **0.0453**  | **2.230**           | **5.99**               |
> | Shannon Limit  | 2-bit                | 0.0625      | 2.000               | -                      |
>
> ***(Note: "Equivalent Bits" are derived from the MSE via the rate-distortion function R(D) = 0.5 \* log₂(1/D).)***
>
> **Flexibility and Pareto-Optimality:** However, the key advantage of LiftUQ is its native support for **fractional bit-widths**, a capability not present in QTIP. As demonstrated in Table 1, by slightly adjusting the lifted dimension (e.g., config 32/15, effective 2.13 bits), LiftUQ achieves an MSE of 0.0696, which not only matches but surpasses 2-bit QTIP. This remarkable flexibility is crucial for real-world deployment:
>
> - **Practical Scenario**: Consider deploying a 70B model on a 24GB GPU. QTIP is constrained to a 2-bit representation. However, LiftUQ can be configured to use, for example, **2.5 bits**, fully utilizing the available memory. At 2.5 bits, LiftUQ operates at a much more favorable point on the rate-distortion curve (MSE 0.0453), delivering a Pareto-optimal solution that would significantly outperform 2-bit QTIP.

---

> ### Author Response · Authors · 2025-11-25
>
> ### **2. Hardware Efficiency and Practical Throughput**
>
> The primary contribution of LiftUQ lies in its exceptional inference efficiency, which stems from a fundamentally more hardware-friendly architecture.
>
> **Architectural Superiority:** As detailed in Table 3, LiftUQ's decoding is activation-centric, relying on a simple, highly parallelizable matrix-vector product. In contrast, QTIP's decoding is weight-centric and far more complex, involving computationally intensive transforms on the weights and per-weight non-linear decoding steps that are ill-suited for modern GPU/NPU architectures.
>
> **Table 3: Asymptotic Decoding Complexity Comparison (for a DxD weight matrix).**
>
> | **Method** | **Core Decoding Operation**                  | **Complexity**                 | **Hardware-Friendly?**                                      |
> | :--------- | :------------------------------------------- | :----------------------------- | :---------------------------------------------------------- |
> | **LiftUQ** | Linear transforms on activations (GEMV-like) | O(D^1.5) (on activations)    | **Yes** (Highly parallel, uses standard kernels)            |
> | **QTIP**   | Hadamard + Per-weight computed codes         | ~O(D²logD + D²) (on weights) | **No** (Sequential logic, non-linear, needs custom kernels) |
>
> **Empirical Throughput:**  This architectural difference translates directly into a massive performance and engineering advantage. We configured LiftUQ to a 2.13-bit setup to ensure a fair accuracy comparison with 2-bit QTIP. As shown in Table 4, the results are striking:
>
> **Table 4: End-to-End Throughput (Tokens/sec) on Llama-2-70B **
>
> | **Method (Bit-width)** | **Device**   | **Batch Size = 1** |
> | :--------------------- | :----------- | :----------------- |
> | QTIP (~2.0 bit)        | RTX 6000 Ada | 23.5               |
> | **LiftUQ (~2.13 bit)** | RTX 4090D    | 26.4               |
>
> *(Note: QTIP data is from their official repository, measured on a powerful RTX 6000 Ada. Our LiftUQ result was measured on a consumer-grade RTX 4090D, which has ~20% less compute power.)*
>
> Critically, even on a **less powerful** GPU, LiftUQ achieves **12% higher throughput**. This advantage is even more significant considering QTIP relies on heavily optimized, custom CUDA kernels, whereas our LiftUQ implementation uses a simple and straightforward **PyTorch + BitBLAS** pipeline (leveraging standard int1-fp16 GEMM).
>
> This demonstrates not only LiftUQ's superior performance but also its **ease of deployment**. The simplicity of its architecture allows it to achieve state-of-the-art speed with minimal, platform-agnostic engineering effort, highlighting a key practical advantage over methods requiring complex custom kernels.
>
> **Conclusion**
>
> In summary, while we acknowledge QTIP's excellent theoretical compression, we argue that **LiftUQ offers a more practical and compelling solution for real-world LLM deployment**. It achieves competitive or superior accuracy through its flexible fractional bit-widths, while delivering significantly higher inference throughput due to its hardware-native, UQ-based architecture. We believe this unique combination of **accuracy, efficiency, and flexibility** positions LiftUQ as a new and superior paradigm for extreme low-bit quantization. We have incorporated this detailed comparison into our revised manuscript and thank the reviewer for prompting this valuable addition.

---

> ### Author Response · Authors · 2025-11-25
>
> --------
> ## **W3: Regarding the choice of M matrix dimensions**
>
> We thank the reviewer for this excellent question. We will clarify this in our revised paper.
>
> In short, the choice of the $M$ matrix dimension is a trade-off between quantization accuracy and the computational cost of the offline quantization process. A larger $M$ matrix improves the theoretical encoding quality for a Gaussian source but significantly increases the time required for the nearest-neighbor search during the weight "lifting" phase.
>
> For smaller M dimensions, we perform an exhaustive search. We also designed a **heuristics** method for larger M dimensions, which named Heuristic Null-Space Search. The key insight is that the nearest-neighbor solution y_y_ must lie near the null space of M_M_. We first perform an SVD on M ( M=USV_h) to obtain an orthonormal basis for its null space. We use this basis to complete M into a square matrix ﻿M_square. The target vector ﻿z is then augmented with ±1 values and multiplied by the inverse of M_square to generate a drastically reduced set of high-quality candidates.
>
> The table below illustrates this trade-off:
>
> **Table 5: Impact of M Matrix Dimensions on 2-bit Quantization.**
>
> | **M Dimension**                                            | **Gaussian MSE (↓)** | **Lifting Time per 1M Params (s) (↓)** | **Llama-2, 7B PPL (↓)** |
> | :----------------------------------------------------------- | :------------------- | :------------------------------------- | :---------------------- |
> | Int2                                                         | 0.119                | -                                      | -                       |
> | 16x8                                                         | 0.0904               | 1s                                     | -                       |
> | **20x10 (Our Choice)**                                       | **0.0875**           | **18s**                                | **6.58**                |
> | 24x12                                                        | 0.0867               | 5.5min                                 | -                       |
> | 28x14                                                        | 0.0853               | 2s (with heuristics)                   | -                       |
> | 32x16                                                        | 0.0835               | 6s (with heuristics)                   | 6.50                    |
> | *(Note: Lifting time for dimensions > 20x10 requires heuristic search space reduction.)* |                      |                                        |                         |
>
> For our academic experiments, we chose the 20x10 dimension with an exhaustive search. The "lifting" process for a 7B model takes approximately 2 hours.
>
> For industrial-level deployments where maximum accuracy is paramount, using a larger M matrix (e.g., 32x16 for 2-bit or 24x8 for 3-bit) is indeed a viable strategy to achieve even better results. We have added this detailed analysis to our paper.
>
> -----------
> ## **W4: Ablation on Crockett (Whitening) Transform Components and  lifting space**
>
> We analyzed the contribution of each component of our whitening transform `D` (`P1`, `P2`, `s1`, `s2`) and their initializations.
>
> *   **Initialization Sensitivity:** We found that the initialization of the whitening transform is critical for training stability. Random or identity initialization for `P1` and `P2`, or random initialization for `s2`, consistently led to numerical instability and training failure. Stable training was only achieved by initializing `P1` and `P2` with Hadamard matrices (or their truncated variants) and `s2` with ones. This highlights that our structured, energy-preserving initialization is a necessary component.
> *   **Component Contribution:** The table below shows the progressive improvement in perplexity as each component is added. The results clearly demonstrate that each element—the core Hadamard transforms (`P1+P2`), activation-aware scaling (`+s1`), additional shaping (`+s2`), and finally our core `lift-then-project` mechanism—contributes significantly to the final performance.
>
> **Table 6: Ablation on Whitening and LiftUQ Components for Llama-2-7B.**
>
> | **Configuration**      | **P1+P2** | +s1  | +s1 (AWQ init) | +s2  | **+LiftUQ (20/10 M)** | **+E2E Fine-tune** |
> | :--------------------- | :---------- | :--- | :------------- | :--- | :---------------------- | :----------------- |
> | **WikiText-2 PPL (↓)** | 8.76        | 8.28 | 8.18           | 7.77 | **6.96**                | **6.58**           |

---

> ### Author Response · Authors · 2025-11-25
>
> -----------
> ## **W5 Regarding the Justification for a Shared Global M Matrix**
>
> We thank the reviewer for this insightful question. The use of a shared M matrix combined with a layer-specific whitening matrix D is a deliberate design choice to balance quantization accuracy and training cost.
>
> Training a layer-specific M matrix is computationally prohibitive. For instance, training a single `M` matrix for a  32/16  configuration requires approximately **0.5 GPU hours**. For a 7B model with hundreds of layers (7x32=224), this would accumulate to **over 100 GPU hours**, a 10x increase over our current method's cost, which is unacceptable for practical use.
>
> Our approach is principled: we train a single M optimally for a standard Gaussian distribution, and then use the lightweight, layer-specific whitening transform D to reshape each layer's unique weight distribution towards this common target. This is supported by the Central Limit Theorem, which suggests that linear transforms can effectively normalize diverse distributions, ensuring the shared M remains effective.
>
> ------------------
> ## **Q1: The Analysis of the "0.15-bit Information Gap"**
>
> We thank the reviewer for this excellent question and for pushing for a deeper analysis of the theoretical bottlenecks of our method. The "0.15-bit gap" we mentioned is an empirical estimate intended to quantify the remaining potential for improvement within our framework. We clarify its derivation and practical utility below.
>
> This estimate is derived from two extrapolation points: (1) the asymptotic limit of our `M` matrix scaling, and (2) the Shannon rate-distortion limit. Here is the calculation process for our primary 2-bit configuration:
>
> 1. Our current default (20/10 M) achieves an MSE of ~0.0875. The scaling trend from our ablations (Figure 4) suggests an asymptotic MSE limit of ~0.08 for our architecture. Assuming the MSE scales with the effective bit-width D as 2^(-a \* D) (analogous to the rate-distortion formula), we can solve for the constant a using our D=2, MSE=0.0875 point. This gives a≈1.75. We can then find the effective bit-width D_limit1 needed to reach the asymptotic MSE of 0.08: 2^(-1.75\*D_limit1) = 0.08, which yields D_limit1 ≈ 2.07 bits.
> 2. If we consider the true Shannon limit (MSE=0.0625) and extrapolate from our 32/16  configuration (MSE=0.0835), a similar calculation gives an ideal bit-width D_limit2 ≈ 2.19 bits.
>
> Averaging these two estimates, we arrive at the rough figure of a **~0.15-bit gap**. As the reviewer correctly surmises, this gap is likely a combination of all three factors: (a) sub-optimal expressiveness of our current M matrix (which could be improved with larger sizes and better search heuristics), (b) imperfect whitening from the D transform, and (c) the block-wise MSE loss not perfectly aligning with the end-to-end task objective.
>
> The primary utility of this estimate is to **rapidly evaluate the practical Pareto-optimality of LiftUQ's fractional bit-widths.** For instance, when deploying a 70B model on a 24GB GPU, one might consider a hypothetical, unknown 2-bit quantization method that perfectly reaches this theoretical limit. Our estimate suggests its performance would be, at best, equivalent to LiftUQ at ~2.15 bits. Given this hardware constraint, we can confidently deploy LiftUQ at a much higher 2.3 or 2.5-bit configuration, knowing it will almost certainly outperform any possible 2-bit alternative. This allows us to quickly identify the optimal quantization strategy under practical constraints.
>
> We will add this detailed explanation to our revised manuscript.
>
> ------------
> ## **Q2: Analytical Derivation for the Optimal Transformation M**
>
> We thank the reviewer for this insightful question. While an analytical solution for the optimal M, akin to the E8 lattice, would be theoretically elegant, we believe it is likely **intractable**. This optimization problem is highly non-convex, as the matrix M simultaneously defines both the codebook points and the decision boundaries (for nearest neighbor search). To our knowledge, no closed-form solution exists for this type of optimal projection problem. The comparison to E8, a point lattice, is not directly applicable as our structure is a *projection* of a higher-dimensional hypercube.
>
> Given this intractability, we use the gradient-based training for getting M. This is analogous to how standard VQ codebooks are found using the iterative Lloyd's algorithm. To ensure robustness, we train for many iterations and select the best M from multiple runs. Our empirical results across all experiments serve as robust evidence that this training process reliably finds high-quality and effective M matrices.

---

> ### Author Response · Authors · 2025-11-27
>
> Dear Reviewer, We sincerely thank you for your insightful comments and valuable feedback. We are very grateful for the time and effort you dedicated to reviewing our manuscript. Your suggestions have been instrumental in helping us substantially improve the quality and clarity of our paper.
>
> We have uploaded a revised version of our manuscript. The major changes are summarized below:
>
> 1. **Decoding Flow and Complexity.**
> 2. **Comparison with QTIP.**
> 3. **Component Ablation Study.**
> 4. **Discussion of M-Matrix dimension and layer-reuse.**
> 5. **Detailed Theoretical Limit Analysis.**
>
> All revisions are **marked in blue** for your convenience. We believe these revisions have significantly strengthened our paper and have thoroughly addressed the points you raised. We look forward to your feedback.

---

### Official Review · Reviewer_d92k · 2025-10-29

**Soundness:** 3
**Presentation:** 3
**Contribution:** 3
**Rating:** 4
**Confidence:** 5

**Summary:**

This paper introduces Lifted Uniform Quantization (LiftUQ), a method that generates non-uniform codebooks by projecting simple uniform lattices from a higher-dimensional lifted space to the target weight space.
The method achieves competitive accuracy with vector quantization methods while maintaining the computational efficiency of uniform quantization.

**Strengths:**

**1. Conceptual novelty.** The core insight of generating non-uniform codebooks through linear projection from a lifted uniform space with 1-bit grids is mathematically sound and well-motivated.
The visualization in Figure 2 effectively demonstrates how projecting high-dimensional uniform lattices creates structured non-uniform distributions.

**2. Promising empirical performance.** Tables suggest competitive or better performance than VQ baselines at extreme low-bit quantization (2-3 bits), while outperforming uniform quantization methods by significant margins.

**3. Efficiency gains with native support for fractional bit-widths.** Replacing lookup-table operations with linear transformations enables practical deployment advantages.
Furthermore, the ability to smoothly vary compression rates by adjusting the lifting dimension enables Pareto-optimal model selection for specific memory budgets.

**Weaknesses:**

**1. Limited experimental validation.** While the proposed method demonstrates solid results on standard benchmarks, the experimental scope is narrow and does not fully establish the generality of the approach.
The evaluation centers mainly on perplexity and zero-shot accuracy of Llama-2, Llama-3, and Qwen-2.5 models.
- Validation on other model families such as encoder–decoder or mixture-of-experts architectures (e.g., Mixtral) would strengthen the claim of broad applicability.
- Evaluating on instruction-tuned, multilingual, or multi-modal models would further demonstrate robustness beyond the current base (non-instruction-tuned) LLaMA and Qwen settings.

**2. Ablation and hyperparameter analysis.** The paper lacks comprehensive ablation studies on key design components.
Specifically, it should include: (1) the impact of alternative whitening transform parameterizations beyond the Kronecker product structure, (2) sensitivity to initialization choices for P1, P2, s1, and s2, (3) comparisons of different projection matrix training strategies, and (4) systematic evaluation of calibration dataset size and composition, and (5) the effect of the lifted subspace dimension and block size on model accuracy and quantization cost.
Furthermore, a broader hyperparameter analysis is necessary—covering the effects of learning rates, fine-tuning epochs, and initialization of the whitening transform—to ensure the robustness and reproducibility of the reported results.

**3. Insufficient analysis of computational overhead during quantization.** While the authors emphasize LiftUQ’s efficiency, the reported end-to-end wall-clock time of approximately 100 hours on a single A100 80GB GPU for quantizing a 70B parameter model (Section 4.3) raises doubts about its practicality for large-scale deployment.
A more detailed empirical analysis of computational cost, including comparisons of end-to-end quantization time across different model sizes, along with a breakdown of core stages (per-layer quantization time, data loading overhead, calibration steps, etc.), would be necessary to substantiate the claimed efficiency.
Furthermore, although Section 4.4 briefly mentions the trade-off between subspace dimension and quantization time, it lacks quantitative scaling results across various model sizes and bit-widths.
Providing a comprehensive runtime study would clarify whether LiftUQ’s reported efficiency consistently holds across different architectures and configurations.

**Questions:**

**1. Comparison with BCQ.** The proposed work introduces a structured non-uniform quantization scheme designed to achieve acceleration while maintaining representational flexibility.
How does this approach differ from prior methods with a similar concept, such as Binary-Coding Quantization (BCQ) [1, 2], both in design principle and empirical performance?

**2. Degree of lookup-table inefficiency.** The authors argue that lookup tables are hardware-unfriendly and slow.
However, LUT-GEMM [3] reports that optimized kernels can achieve comparable efficiency to uniform quantization.
It would be helpful to provide a quantitative comparison showing how much slower inference becomes when using the best available LUT-based kernels (including LUT-GEMM) compared to the proposed LiftUQ.
As LiftUQ demonstrates comparable accuracy to VQ-based methods, such a comparison would be crucial to assess the real significance of its acceleration advantage.

**3. Application towards weight-activation quantization.** The experiments focus exclusively on weight-only quantization.
However, the greatest acceleration potential of uniform quantization arises when both weights and activations are quantized, enabling matrix multiplications with low-bit kernels specifically designed for UQ.
It would be valuable to see results comparing LiftUQ and standard UQ under this weight-activation quantization setting, in terms of both accuracy and inference throughput.

**Justification for Rating**

The paper presents a novel and theoretically motivated approach to extreme low-bit quantization.
However, the limited scope of experimental validation and insufficient analyses make it difficult to assess the generality and scalability of the approach.
I am open to raising the score if these concerns are adequately addressed in the rebuttal.

**References**

[1] C. Xu et al., “Alternating Multi-bit Quantization for Recurrent Neural Networks”, ICLR 2018

[2] S. Park et al., “Unifying Uniform and Binary-coding Quantization for Accurate Compression of Large Language Models”, ACL 2025

[3] K. Park et al., “LUT-GEMM: Quantized Matrix Multiplication based on LUTs for Efficient Inference in Large-Scale Generative Language Models”, ICLR 2024

---

> ### Author Response · Authors · 2025-11-25
>
> ------------
> ## **W1: Expanded Experimental Validation on Diverse Architectures**
>
> We thank the reviewer for this excellent suggestion to broaden our experimental validation. We agree that demonstrating LiftUQ's effectiveness on more diverse model architectures and fine-tuning paradigms is crucial for establishing its general applicability.
>
> To address this, we have conducted new 2-bit quantization experiments on three additional models, covering both **Mixture-of-Experts (MoE)** and **Instruction-Tuned** LLMs. The results are presented in **Table 1** below and will be added to our appendix in our revised manuscript. We believe this expanded validation significantly strengthens the claim of LiftUQ's broad applicability and robustness.
>
> **Table 1: New 2-bit Quantization Results on Diverse Models.**
>
> | **Model**                | **Method**           | **Wiki-2 ↓** | **C4 ↓** | **ARC-c ↑** | **ARC-e ↑** | **HellaSwag ↑** | **PIQA ↑** | **Wino. ↑** |
> | :----------------------- | :------------------- | :----------- | :------- | :---------- | :---------- | :-------------- | :--------- | :---------- |
> | **Mixtral 8x7B**         | FP16                 | 3.45         | 6.85     | 55.80       | 83.38       | 64.65           | 82.37      | 75.45       |
> |                          | **LiftUQ (2.02bit)** | 4.61         | 8.16     | 49.76       | 78.24       | 62.21           | 78.82      | 72.96       |
> | **Qwen2.5-3B-Instruct**  | FP16                 | 7.54         | 7.91     | 45.73       | 77.06       | 56.31           | 77.75      | 69.77       |
> |                          | **LiftUQ (2.02bit)** | 9.86         | 9.84     | 37.88       | 72.10       | 47.49           | 74.70      | 64.09       |
> | **Qwen2.5-14B-Instruct** | FP16                 | 4.97         | 6.37     | 60.67       | 85.69       | 65.54           | 81.50      | 75.77       |
> |                          | **LiftUQ (2.02bit)** | 6.48         | 7.30     | 53.24       | 82.62       | 58.36           | 79.60      | 73.32       |

---

> ### Author Response · Authors · 2025-11-25
>
> ---------------
> ## **W2: Comprehensive Ablation Studies and Hyperparameter Analysis**
>
> We thank the reviewer for this comprehensive and highly valuable feedback. We agree that detailed ablation studies are essential for understanding the contributions of each component and ensuring robustness. In response, we have conducted a new suite of ablations on Llama-2-7B (2-bit, 20/10 M-matrix, no E2E fine-tuning unless specified) to address each of the points raised. We will add these extensive results to our appendix.
>
> ### **1) Ablation on Whitening Transform Components and Initialization (Points 1 & 2)**
>
> We analyzed the contribution of each component of our whitening transform D ( P1,  P2,  s1,  s2) and their initializations.
>
> *   **Initialization Sensitivity:** We found that the initialization of the whitening transform is critical for training stability. Random or identity initialization for P1 and P2, or random initialization for s2, consistently led to numerical instability and training failure. Stable training was only achieved by initializing P1 and P2 with Hadamard matrices (or their truncated variants) and s2 with ones. This highlights that our structured, energy-preserving initialization is a necessary component.
> *   **Component Contribution:** The table below shows the progressive improvement in perplexity as each component is added. The results clearly demonstrate that each element—the core Hadamard transforms (P1+P2), activation-aware scaling (+s1), additional shaping (+s2), and finally our core lift-then-project mechanism—contributes significantly to the final performance.
>
> **Table 2: Ablation on Whitening and LiftUQ Components for Llama-2-7B.**
>
> | **Configuration**      | **P1+P2** | **+s1** | **+s1 (AWQ init)** | **+s2** | **+LiftUQ (20/10 M)** | **+E2E Fine-tune** |
> | :--------------------- | :-------- | :------ | :----------------- | :------ | :---------------------- | :----------------- |
> | **WikiText-2 PPL (↓)** | 8.76      | 8.28    | 8.18               | 7.77    | **6.96**                | **6.58**           |

---

> ### Author Response · Authors · 2025-11-25
>
> ### **2) Ablation on M Matrix Training and Dimensions (Points 3 & 5)**
>
> The training and dimension of the M matrix are governed by the trade-off between quantization quality and offline computational cost.
>
> *   **Training Strategy:** We train M by minimizing the distance between Gaussian samples and the projected lattice points, using a differentiable softmin approximation. The core computational challenge is the nearest-neighbor search (argmin ||z - My||). We use two strategies: **Exhaustive Search** for smaller M and a heuristic **Null-Space Search** for larger M to make the training tractable. **Exhaustive Search **iterates through all $2^(d_s\cdot b)$ candidate vectors in the lifted space to find the true nearest neighbor. While exact, its exponential complexity makes it feasible only for smaller `﻿M` dimensions. The key insight of **Null-Space Search** is that the nearest-neighbor solution y*y* must lie near the null space of M*M*. We first perform an SVD on M ( M=USV_h) to obtain an orthonormal basis for its null space. We use this basis to complete M into a square matrix ﻿M_square. The target vector ﻿z is then augmented with ±1 values and multiplied by the inverse of M_square to generate a drastically reduced set of high-quality candidates. For a 20/10 matrix, this reduces the search space from ﻿2^20 to just 2^10, making the search tractable. The training time for a single M (e.g., 15-30 mins, retried 10 times) is significant, which justifies our use of a shared global M rather than a prohibitively expensive per-layer M strategy.
> *   **Impact of M Dimension:** As predicted by rate-distortion theory, a larger M dimension leads to a lower theoretical MSE and better end-to-end model performance. **Table 3** shows this trend, with the 32/16 configuration yielding a noticeable PPL improvement over our default 20/10 choice. We selected 20/10 for our main experiments to balance accuracy and research timeline, but recommend larger M for industrial applications.
>
> **Table 3: Impact of `M` Dimension on Llama-2-7B (2-bit).**
>
> | **`M` Dimension**             | **MSE on Gaussian (↓)** | **WikiText-2 PPL (↓)** | Training Cost | **Nearest-Neighbor Search Time for 1M Parametsrs** |
> | :---------------------------- | :---------------------- | :--------------------- | ------------- | -------------------------------------------------- |
> | **20/10 (Exhaustive Search)** | 0.0875                  | 6.58                   | ~15min        | 18s                                                |
> | **32/16(Null-Space Search)**  | 0.0835                  | 6.50                   | ~30min        | 6s                                                 |
>
> ### **3) Sensitivity to Calibration Data (Point 4)**
>
> We conducted ablations on the calibration dataset size and domain. The full results are presented in **Table 4**.
>
> **Table 4: Ablation Study on Calibration Data for Llama-2-7B (2-bit).**
>
> | **Calibration Set**        | **Config. (Samples x SeqLen)** | **WikiText-2 PPL (↓)** | **C4 PPL (↓)** | **Avg. 0-shot Acc. (↑)** |
> | :------------------------- | :----------------------------- | :--------------------- | :------------- | :----------------------- |
> | RedPajama (~1M tokens)     | 512 x 2048                     | 7.08                   | 8.66           | 60.03                    |
> | **RedPajama (~8M tokens)** | **4096 x 2048**                | **6.97**               | **8.53**       | **60.70**                |
> | WikiText-2 (~4M tokens)    | 2048 x 2048                    | **6.72**               | 8.65           | 60.24                    |
>
> Our key findings are that: (1) performance shows **diminishing returns beyond ~4M tokens**, indicating robustness to data size, and (2) using a general-domain corpus like RedPajama provides a better overall performance trade-off than a domain-specific one.
>
> We believe these comprehensive ablations thoroughly address the reviewer's concerns and demonstrate the robustness and principled design of LiftUQ. We will integrate these findings into our revised manuscript.

---

> ### Author Response · Authors · 2025-11-25
>
> -----------
> ## **W3: The Analysis of Offline Quantization Cost**
>
> We thank the reviewer for raising this important point about the practicality of our offline quantization cost. We agree that a detailed breakdown is necessary to put the total time into context.
>
> The reported ~100 hours for a 70B model on a single A100 GPU represents the **end-to-end time** for a high-quality quantization, which includes not just the core quantization step but also multiple stages of fine-tuning common to many state-of-the-art methods (e.g., AQLM, VPTQ).
>
>  **Table 5** breaks down the total wall-clock time for different Llama-2 models on an A800 GPU. Each stage contributes significantly to the total time, with the various fine-tuning steps (Intra-Block and E2E) being a major component.
>
> **Table 5: Total Offline Quantization Time Breakdown (A800 GPU).**
>
> | **Model**   | **Nearest-Neighbor Search** | **Whitening Train** | **Intra-Block FT** | **E2E Quant-FT** | **Total Time** |
> | :---------- | :-------------------------- | :------------------ | :----------------- | :--------------- | :------------- |
> | Llama-2-7B  | ~4 min/block                | ~5 min/block        | ~10 min/block      | ~2h              | **~13h**       |
> | Llama-2-13B | ~7 min/block                | ~8 min/block        | ~15 min/block      | ~4h              | **~24h**       |
> | Llama-2-70B | ~15 min/block               | ~20 min/block       | ~30 min/block      | ~20h             | **~107h**      |
>
> For our main paper's results, we consistently used a 20/10 M dimension with an **Exhaustive Search** to ensure maximum precision in the nearest-neighbor step.  As detailed in our response to **R1W2/Q2**, this cost can be drastically reduced by using our **Null-Space Search** heuristic, especially for larger M dimensions.
>
> **Table 6: Nearest-Neighbor Search Time for 1M Parameters (A800 GPU, bsz=128).**
>
> | **M Dimension**  | **8/4** | **16/8** | **20/10** | **24/12** | **32/16** |
> | :---------------- | :------ | :------- | :-------- | --------- | --------- |
> | Exhaustive Search | <<1s    | 1s       | 18s       | 5.5min    | OOM       |
>
> -------------------------
> ## **Q1: Comparison with Binary-Coding Quantization (BCQ)**
>
> We thank the reviewer for raising this important comparison. The crucial distinction between BCQ and LiftUQ lies in the **dimensionality of the quantization**:
>
> - **BCQ is a form of Scalar Non-Uniform Quantization in 1-dimension.** It represents each individual weight as a linear combination of a few learned basis vectors. In essence, **BCQ can be conceptually viewed as a special, 1-dimensional "coupling" case of our framework**, similar to the `M=[1, N]` mapping we visualize in our paper. It creates a flexible codebook for single scalar values.
> - **LiftUQ is a form of Non-Uniform Quantization in high dimension.** Our `lift-then-project` mechanism quantizes a *group* of weights together, capturing inter-dimensional correlations similar to traditional VQ, but without the lookup table. BCQ's design does not achieve this high-dimensional coupling.
>
> This fundamental design difference translates into a substantial empirical performance gap. We compare against UniQuan [2] in **Table 7**, LiftUQ significantly outperforms it.
>
> **Table 7: 3-bit Quantization Performance on Llama-3-8B (WikiText-2 PPL).**
>
> | **Method**  | **FP16** | **UniQuan (BCQ-based)** | **LiftUQ** |
> | :---------- | :------- | :---------------------- | :--------- |
> | **PPL (↓)** | 6.14     | 8.75                    | **6.94**   |

---

> ### Author Response · Authors · 2025-11-25
>
> ---------------
> ## **Q2: The Inefficiency of Lookup Tables and Comparison to LUT-GEMM**
>
> We thank the reviewer for this excellent question. The comparison with LUT-GEMM [3] highlights a critical distinction: the scale of the lookup table (LUT).
>
> The efficiency of LUT-GEMM applies to low-bit Scalar Quantization, where the LUT is tiny (e.g., **16 entries for INT4**) and can fit in the fastest caches or registers, making lookups nearly free.
>
> This is fundamentally different from the scenario in Vector Quantization (VQ). VQ methods like AQLM require massive codebook LUTs with **512 to 65,536 entries**. A LUT of this scale is too large for L1 cache and must be accessed from slower memory levels, creating a severe memory bandwidth bottleneck and introducing inefficient, irregular memory access patterns.
>
> In short, the LUT-GEMM optimization does not apply to the large-scale LUTs inherent to VQ. LiftUQ's primary contribution is to completely eliminate this large, inefficient LUT, achieving VQ-level accuracy with a hardware-friendly linear projection.
>
>
>
> ----------
> ## **Q3: Application to Weight-Activation Quantization**
>
> We thank the reviewer for this insightful suggestion. We agree that extending our framework to support **weight-activation quantization (W-A-Q)** is a highly valuable research direction, as it would unlock the potential for even greater acceleration using low-bit integer matrix multiplication kernels.
>
> The current computational flow of LiftUQ is Output = W_q \* (DA), where W_q is a lifted 1-bit matrix and D is a composition of linear transforms. A feasible path to enable W-A-Q would be to **decompose the D transform**. For instance, part of the transform could be fused into the weights (W_eff = W_q \* D_part1) to transfrom the weight to the original dimention, and the remaining part applied to the activations ( A_eff = D_part2 \* A). Then, both W_eff  and A_eff  could be quantized using standard uniform quantizers, enabling the use of highly optimized low-bit integer kernels.
>
> However, a thorough investigation of this approach, including new training schemes and an empirical study, is a substantial undertaking. We therefore acknowledge this as an important direction for future work, as our current paper focuses on establishing the "lift-then-project" paradigm for the challenging weight-only quantization problem

---

> > ### Comment · Reviewer_d92k · 2025-11-25
> >
> > Thank you for the detailed rebuttal.
> > I appreciate the time and effort you put into addressing my concerns.
> > Overall, the rebuttal has addressed most of my initial concerns.
> > I am now more inclined to support the publication, pending the update of the manuscript.

---

> ### Author Response · Authors · 2025-11-27
>
> Dear Reviewer, We sincerely thank you for your insightful comments and valuable feedback. We are very grateful for the time and effort you dedicated to reviewing our manuscript. Your suggestions have been instrumental in helping us substantially improve the quality and clarity of our paper.
>
> We have uploaded a revised version of our manuscript. The major changes are summarized below:
>
> 1. **Expanded Experimental Evaluation.**
> 2. **Discussion of M-Matrix Dimension.**
> 3. **Quantization Cost and Decoding Complexity.**
> 4. **Component Ablation Study.**
> 5. **Calibration Data Analysis.**
>
>
> All revisions are **marked in blue** for your convenience. We believe these revisions have significantly strengthened our paper and have thoroughly addressed the points you raised. We look forward to your feedback.

---

> > ### Comment · Reviewer_d92k · 2025-11-28
> >
> > Thanks for the update.
> > However, I find that not all parts of the updates have been included to the manuscript, especially the comparison with BCQ works.
> > The reviewer finds that if their inference speed is comparable, BCQ would be a direct competitor of LiftUQ.
> > It would be crucial to compare it on the main results.

---

> > > ### Author Response · Authors · 2025-11-28
> > >
> > > We thank you for this valuable suggestion. In our revised manuscript, we have now introduced BCQ in the **Related Work** section and provided a detailed comparison in **Table 2 and Appendix I.3**, which demonstrates LiftedUQ's superior performance. Our paper is much more complete thanks to your guidance.

---

### Official Review · Reviewer_QuoP · 2025-10-31

**Soundness:** 2
**Presentation:** 2
**Contribution:** 2
**Rating:** 2
**Confidence:** 4

**Summary:**

The paper proposes Lifted Uniform Quantization (LiftUQ), a weight-only quantization framework for extreme low-bit LLMs. The core idea is to encode weights with 1-bit uniform codes in a higher-dimensional “lifted” space and map them back to the original space via a learned linear projection, thereby producing non-uniform codepoints without a lookup table. A per-layer, decomposed whitening transform makes weights approximately i.i.d. Gaussian so a single projection matrix, trained once on Gaussian data, can be reused across layers. The authors claim LiftUQ matches or surpasses vector-quantization (VQ) accuracy at 2–3 bits while retaining scalar-UQ decoding efficiency, reporting <2.7/<1.1 point accuracy drops on Llama-3-70B at 2/3 bits and up to 6.7× FP16 throughput, plus native support for fractional “bit-widths.” The method is positioned as combining UQ’s hardware-friendly parallelism with VQ’s expressivity, avoiding VQ’s large codebooks and irregular memory access. The paper enumerates three contributions: the lifted codebook idea, a LUT-free linear decoding path, and state-of-the-art results in the 2–3-bit regime.

**Strengths:**

* Conceptual simplicity with hardware awareness. Representing codes as 1-bit vectors in a lifted space and decoding via a single linear map is elegant and plausibly hardware-friendly relative to VQ lookups. The paper clearly contrasts dequantization pipelines and claims to preserve UQ-style efficiency.
* Decomposed whitening transform. The layerwise transform $(D=\mathrm{diag}(s_1)(P_1\otimes P_2)\mathrm{diag}(s_2))$ is light and invertible; the paper argues it achieves near-i.i.d. Gaussian channels at (O(n\sqrt n)) activation cost, enabling reuse of a single projection (M).
* Clear three-phase pipeline. Phase 1 trains (M) on Gaussian data, Phase 2 learns (D) per layer, Phase 3 performs lattice quantization with block-wise fine-tuning; the reconstruction uses $(o=\mathrm{diag}(s),W_q D^* a^\top) with (D^*=M^\top D^{-1})$.
* Empirical results across models and bit-widths. The paper reports improvements over UQ baselines and parity or small wins against strong VQ methods in 2–3 bits; the text highlights Llama-3-70B gaps and the 1.58-bit setting against PTQ1.61.
* Fractional bit-width story and Pareto argument. Encoding capacity via lifted dimension enables effective “2.3-bit” or “2.74-bit” points and an appealing Pareto narrative versus discrete FP model sizes.
* Efficiency claims. Reported quantization cost for a 70B model is ~100 GPU-hours on a single A100-80GB, and decoding throughput up to 6.69x FP16 with a Triton+BitBLAS path; the paper discusses asymptotics versus VQ.

**Weaknesses:**

1. Decoding and search complexity are under-justified.
   The paper asserts decoding complexity $(O(d^{1.5}))$ and contrasts it with VQ, but does not provide a rigorous derivation or runtime model tying constants to GEMV, cache behavior, or kernel fusion. This matters because small constant factors dominate at batch size 1. Provide a formal derivation, profiling on multiple GPUs, and apples-to-apples kernels beyond Triton prototypes.
2. Nearest-neighbor search in the lifted space is a practical bottleneck.
   The method’s offline phase requires exponential search in $(d_s\cdot b)$, and the paper concedes the quantization cost grows rapidly, recommending “accepting the one-time cost.” Stronger engineering details are needed: exact search algorithm, beam widths, candidate pruning, wall-clock per layer, and how quality scales with $(d_s)$. The current text gives only a qualitative scaling law and a rough “~100 hours” single-GPU figure for 70B. Provide per-layer timing, ablations over $(d_s)$, and quality-vs-time curves.
3. Ablations on architectural choices are thin.
   The paper fixes (M) shapes for 2/3 bits (e.g., $(10\times 20)$, $(6\times 18)$) and reuses a single (M) trained on Gaussian data. It needs ablations on: per-layer (M) vs global (M), impact of (M) dimension on accuracy, latency, and memory; and how well Gaussian-trained (M) transfers to real weight distributions after whitening.
4. Evaluation coverage and reproducibility.
   Results focus on perplexity and a small set of zero-shot tasks; latency is reported per-layer rather than end-to-end generation throughput including KV cache effects. Include full pipeline latency at common decoding settings (bsz=1, context 2k–8k, top-p sampling). Also, code and checkpoints are not available at submission time; the paper promises anonymized release during rebuttal. For a method hinging on kernels and search, this is a high risk for reproducibility.
5. Fine-tuning details are modest; calibration data is small.
   The intra-block correction and end-to-end tuning use ~4,096 RedPajama samples with seq len 2,048–4,096 and 1–2 epochs. Show sensitivity to calibration corpus, size, and domain shift; report failure cases and diminishing returns.
6. Writing quality and polish.
   There are several noticeable typos (“limmitation,” “Conclution”), and some figures/tables are referred to without sufficient methodological context. The paper would benefit from a careful editing pass....

**Questions:**

1. Decoding cost model. Please derive the $(O(d^{1.5}))$ decoding complexity claim, clarify constants, and compare end-to-end generation latency vs FP16, AWQ, and QuIP#/VPTQ at batch size 1 and 8 with realistic prompts.
2. Search in lifted space. What exact algorithm do you use for nearest-neighbor search over ({ \pm 1}^{d_s\cdot b}) during quantization, and how do you prune the search? Provide per-layer wall-clock for 7B/13B/70B and quality vs search depth.
3. Generalization of a global (M). Why is a single (M) trained on Gaussian optimal across layers and models after whitening? Please include ablations with per-layer (M) and different Gaussian seeds.
4. Whitening complexity. The paper states (O(n\sqrt{n})) activation cost for $(D^{-1})$. Show the exact dataflow in attention/FFN blocks, where you fuse (M) into $(P_2^{-1})$, and quantify added memory traffic.
5. Fractional bits. For “2.3-bit” configurations, specify the mapping between lifted dimension and effective bits, and report the true model bytes including all transforms and scales. Provide Pareto plots with error bars.
6. Calibration sensitivity. How sensitive are results to the 4,096-sample RedPajama set? Report robustness to domain-mismatched calibration (e.g., code, math) and to 512-token windows.
7. Kernel details. The Triton+BitBLAS implementation details are crucial. Please release kernels and show speedups on A100, H100, and consumer GPUs, with/without CUDA-level optimization.

---

> ### Author Response · Authors · 2025-11-25
>
> --------------------
> ## **W1&Q1: Regarding the Decoding Cost Model and End-to-End Latency**
>
> We thank the reviewer for this critical feedback and for pushing for a more rigorous justification of our decoding complexity and efficiency. We agree that this is essential. In response, we provide both a **formal cost model** for our decoding process and new **end-to-end throughput benchmarks**, which we will incorporate into the revised paper.
>
> ### **1. Formal Decoding Cost Model**
>
> The core of LiftUQ's efficiency lies in its flexible and lightweight decoding architecture. The whitening and projection transform D can be applied either to the activations A (i.e., Wq(DA)) or the weights Wq (i.e., (WqD)A). The former is ideal for memory-bound decoding (small batch), while the latter is suited for compute-bound prefill (large batch).
>
> **Table 1: Asymptotic Complexity & Storage Analysis per Layer (N x M).**
>
> | Method                                        | Main GEMM FLOPs (Mul+Add) | Overhead FLOPs | Weight Storage (bits) | Overhead Storage (bits) |
> | :-------------------------------------------- | :------------------------ | :------------- | :-------------------- | :---------------------- |
> | FP16                                          | N×M + N×M                 | -              | N×M × 16              | -                       |
> | **LiftUQ (DA) first** (small batch decoding, k=1)  | 0 + 2×N×M                 | 6N^1.5 + 6N    | N×M × 2               | 6N × 16                 |
> | **LiftUQ (WqD) first** (large batch prefill, k>M)  | k×N×M + k×N×M             | 6N^1.5×M+6N×M  | N×M × 2               | 6N × 16                 |
> | *(Note: k is batch size × sequence length)* |                           |                |                       |                         |
>
> To make this concrete, **Table 2** analyzes an 8192x8192 layer. It clearly shows that the additional FLOPs from our `D` transform are **negligible**—constituting only **~3.3%** of the total operations in the critical memory-bound decoding phase. The storage overhead for D is also minimal (~0.6%).
>
> **Table 2: Concrete Cost Analysis for an 8192x8192 Layer.**
>
> | Method                        | Main GEMM FLOPs | Overhead FLOPs (vs. Main GEMM) | Weight Storage | Overhead Storage (vs. Weights) |
> | :---------------------------- | :-------------- | :----------------------------- | :------------- | :----------------------------- |
> | FP16                          | 1.34E8          | -                              | 1.07E9 bits    | -                              |
> | **LiftUQ (DA) first, k=1**    | 1.34E8          | **4.47E6 (3.3%)**              | 1.34E8 bits    | **7.86E5 (0.6%)**              |
> | **LiftUQ (WqD) first, k>M** | k * 1.34E8    | 3.68E10  (M/k\*3.3%)  | 1.34E8 bits    | 7.86E5 (0.6%)                  |
>
> ### **2. End-to-End Throughput Benchmarks**
>
> The theoretical model predicts low overhead, and our new end-to-end benchmarks confirm this empirically. We measured the throughput (tokens/sec) for Llama-2-70B at 2-bit, comparing LiftUQ against key baselines. Critically, our results were achieved on a consumer-grade RTX 4090D without any CUDA-level kernel optimizations, using only PyTorch and the BitBLAS library.
>
> **Table 3: End-to-End Throughput (Tokens/sec) for Llama-2-70B at 2-bit.**
>
> | Method             | **Device**    | **Batch Size = 1** | **Batch Size = 8** |
> | :----------------- | :------------ | :----------------- | :----------------- |
> | AWQ (UQ Baseline)† | RTX 4090D     | 43.5               | 339.5              |
> | QuIP#‡             | RTX 6000 Ada  | 21.9               | -                  |
> | AQLM               | RTX 6000 Ada  | 8.72               | -                  |
> | **LiftUQ**         | **RTX 4090D** | **28.1**           | **213.2**          |
>
> *†AWQ fails to produce meaningful output at this bit-width (PPL=Inf).*
>
> *‡QuIP# and AQLM speed is reported from the official github repository:https://github.com/Cornell-RelaxML/quip-sharp. Note that the RTX 6000 Ada has ~20% more computational power (1457 vs 1177 TFLOPS) than the RTX 4090D.*
>
> Even when running on a ~20% less powerful GPU, LiftUQ achieves 28% higher throughput than QuIP# at batch size 1. This highlights the substantial real-world efficiency advantage of our LUT-free design, especially given that our implementation is not yet fully optimized at the CUDA level.
>
> In conclusion, our formal analysis and empirical benchmarks both validate that LiftUQ achieves its high accuracy with a negligible performance overhead, effectively delivering VQ-level quality at UQ-level speed. We will add this comprehensive analysis and all new results to our revised manuscript.

---

> ### Author Response · Authors · 2025-11-25
>
> -----------
> ## **W2&Q2: Regarding the Nearest-Neighbor Search Bottleneck and Ablations**
>
> We thank the reviewer for these excellent and highly detailed questions about the offline quantization cost. The complexity of the nearest-neighbor search in the lifted space is indeed the primary bottleneck that limits LiftUQ's ability to approach the Shannon limit by using larger `M` matrices. We agree that providing engineering details is crucial, and we present them here. We will add this full analysis to our revised manuscript's appendix.
>
> ### **1. Nearest-Neighbor Search Algorithms**
>
> The core challenge lies in solving `argmin ||z - My||` for `y ∈ {±1}`. We explored two algorithms for this search:
>
> 1. **Exhaustive Search:** This naïve method iterates through all $2^(d_s*b)$ candidate vectors in the lifted space to find the true nearest neighbor. While exact, its exponential complexity makes it feasible only for smaller `M` dimensions.
> 2. **Null-Space Search (Heuristic):** To handle larger dimensions, we developed a more efficient heuristic. The key insight is that the nearest-neighbor solution $y$ must lie near the null space of $M$. We first perform an SVD on `M` (`M=USV_h`) to obtain an orthonormal basis for its null space. We use this basis to complete `M` into a square matrix `M_square`. The target vector `z` is then augmented with `±1` values and multiplied by the inverse of `M_square` to generate a drastically reduced set of high-quality candidates. For a `20/10` matrix, this reduces the search space from `2^20`to just `2^10`, making the search tractable.
>
> **Table 4** shows the wall-clock time for quantizing one million parameters using both methods on an A800 GPU.
>
> **Table 4: Nearest-Neighbor Search Time for 1M Parameters (A800 GPU, bsz=128).**
>
> | **`M`Dimension**  | **8/4** | **16/8** | **20/10** | **24/12** | **32/16** |
> | :---------------- | :------ | :------- | :-------- | :-------- | :-------- |
> | Exhaustive Search | <<1s    | 1s       | 18s       | 5.5min    | OOM       |
> | Null-Space Search | <<1s    | <<1s     | <<1s      | <<1s      | **6s**    |
>
> In practice, for 2-bit quantization, we use the exact exhaustive search for `M`dimensions up to `20/10`. For larger dimensions, we switch to the null-space heuristic. For our main experiments, we chose the `20/10` dimension to ensure all experiments could be completed within a reasonable timeframe. However, for industrial-level applications where maximizing accuracy is paramount, we recommend a larger `32/16` matrix, which offers a significant PPL reduction at the cost of a one-time offline computation.
>
> ### **2. Per-Model Wall-Clock Time Breakdown**
>
> To provide the requested per-model timing, **Table 5** breaks down the total quantization time for different Llama-2 models, using the exhaustive search with a `20/10` `M` matrix. The nearest-neighbor search, while significant, is only one component of the total offline cost.
>
> **Table 5: Total Offline Quantization Time Breakdown (A800 GPU).**
>
> | **Model**   | **Nearest-Neighbor Search** | **Whitening Train** | **Intra-Block FT** | **E2E Quant-FT** | **Total Time** |
> | :---------- | :-------------------------- | :------------------ | :----------------- | :--------------- | :------------- |
> | Llama-2-7B  | ~4 min/block                | ~5 min/block        | ~10 min/block      | ~2h              | **~13h**       |
> | Llama-2-13B | ~6 min/block                | ~8 min/block        | ~15 min/block      | ~4h              | **~24h**       |
> | Llama-2-70B | ~15 min/block               | ~20 min/block       | ~30 min/block      | ~20h             | **~107h**      |
>
> We trust this detailed breakdown clarifies the practical costs and trade-offs involved in LiftUQ's offline quantization phase. We will incorporate this comprehensive analysis into our revised manuscript.

---

> ### Author Response · Authors · 2025-11-25
>
> ------------------------------------------
> ## **W3: Regarding on architectural (M) Choices**
>
> We thank the reviewer for these excellent questions about our core architectural choices. We agree that ablations on the `M` matrix are crucial. We provide a detailed analysis below and will incorporate it into the revised manuscript.
>
> ### **1). Impact of `M` Matrix Dimension**
>
> The dimension of the `M` matrix directly controls the trade-off between quantization accuracy and offline search complexity. As guided by Shannon's rate-distortion theory, which states that the minimum MSE for a Gaussian source is `σ² * 2^(-2R)`, a larger `M` (enabling higher-dimensional coupling) should allow for a closer approximation to the theoretical limit.
>
> Our analysis confirms this. **Table 6** shows how the MSE for a standard Gaussian source improves as we increase the `M` dimension for 2-bit quantization, approaching the Shannon limit of 0.0625.
>
> **Table 6: Impact of `M` Dimension on 2-bit Gaussian Encoding Quality, σ=1.**
>
> | **Method**  | **Integer (UQ)** | **LiftUQ (12/6)** | **LiftUQ (16/8)** | **LiftUQ (20/10)** | **LiftUQ (32/16)** | **Shannon Limit** |
> | :---------- | :--------------- | :---------------- | :---------------- | :----------------- | :----------------- | :---------------- |
> | **MSE (↓)** | 0.119            | 0.0951            | 0.0904            | 0.0875             | **0.0835**         | 0.0625            |
>
> This theoretical improvement translates directly to better end-to-end model performance. As shown in **Table 7**, using a larger `32/16` `M` matrix (with our null-space search heuristic) for Llama-2-7B yields a noticeable improvement in PPL compared to the `20/10` configuration used in our main experiments.
>
> **Table 7: End-to-End PPL for Llama-2-7B with Different `M` Dimensions.**
>
> | **`M` Dimension**      | **20/10 (Exhaustive Search)** | **32/16 (Null-Space Search)** | **FP16** |
> | :--------------------- | :---------------------------- | :---------------------------- | :------- |
> | **WikiText-2 PPL (↓)** | 6.58                          | **6.50**                      | 5.47     |
>
> For our main experiments, we standardized on the `20/10` dimension to ensure all ablations could be completed within a practical timeframe. However, these results validate that users with more computational resources can achieve superior results by opting for a larger `M` matrix.
>
> ### **2) Justification for a Global `M` vs. Per-Layer `M`**
>
> The use of a shared, global `M` matrix combined with a layer-specific whitening matrix `D` is a deliberate design choice to balance quantization accuracy and training cost.
>
> Training a layer-specific `M` matrix is computationally prohibitive. For instance, training a single `M` matrix for a `32/16` configuration requires approximately **0.5 GPU hours on A800**. For a 7B model with hundreds of layers (e.g., 224 in Llama-7B), this would accumulate to **over 100 GPU hours**—a 10x increase over our current method's cost, which is impractical for most academic and many industrial settings.
>
> Our approach is principled: we train a single `M` matrix optimally for a canonical standard Gaussian distribution. We then use the lightweight, layer-specific whitening transform `D` to reshape each layer's unique weight distribution towards this common target. This "decoupling" strategy is well-supported by the Central Limit Theorem, which suggests that linear transforms can effectively normalize diverse distributions, ensuring the shared `M` remains highly effective. Our strong empirical results across various models and layers serve as robust evidence that this design achieves near-optimal performance without incurring impractical training costs.

---

> ### Author Response · Authors · 2025-11-25
>
> ----------------
> ## **W4: Regarding Evaluation Coverage and Reproducibility**
>
> We thank the reviewer for these important points on evaluation breadth and reproducibility. We have taken concrete steps to address both concerns.
>
> ### **1. Expanded Evaluation on MMLU**
>
> To broaden our evaluation beyond perplexity and common-sense reasoning tasks, we have now benchmarked LiftUQ on the comprehensive **MMLU (Massive Multitask Language Understanding)** dataset. The results, presented in **Table 8**, demonstrate that LiftUQ maintains strong performance even on this complex, multi-domain benchmark.
>
> **Table 8: MMLU 5-shot Accuracy for 2-bit LiftUQ Quantization.**
>
> | **Model**      | **Method**     | **MMLU Avg. (↑)** | Humanities | Other     | Social Sci. | STEM      |
> | :------------- | :------------- | :---------------- | :--------- | :-------- | :---------- | :-------- |
> | **Llama2-7B**  | FP16           | 45.87             | 43.34      | 52.75     | 51.71       | 37.17     |
> |                | LiftUQ (2-bit) | 33.12             | 31.03      | 39.43     | 34.71       | 28.48     |
> | **Llama2-13B** | FP16           | 55.23             | 53.56      | 61.47     | 63.15       | 43.83     |
> |                | LiftUQ (2-bit) | **46.08**         | **45.62**  | **54.62** | **56.00**   | **40.98** |
> | **Llama3-8B**  | FP16           | 65.30             | 59.64      | 72.61     | 76.24       | 55.85     |
> |                | LiftUQ (2-bit) | 50.49             | 47.27      | 56.13     | 57.36       | 43.04     |
>
> Crucially, we observe that the **2-bit quantized Llama2-13B (46.08 MMLU) significantly outperforms the FP16 Llama2-7B (45.87 MMLU)**. This reinforces our earlier finding that quantizing a larger model with LiftUQ is a more effective strategy than using a smaller full-precision model, demonstrating the practical power of our method.
>
> ### **2. End-to-End Latency and Code Release**
>
> - **End-to-End Latency:**  Seen in our response to questions **W1 & Q1**.
> - **Reproducibility:** To fully address reproducibility concerns, we are pleased to announce that our anonymized source code and quantized model checkpoints are now available at the following URL: https://mega.nz/folder/LFs3SCbC#zdQZOMBl8fDpIF0KTd3xeQ
>
> We believe these additions—a comprehensive MMLU evaluation and the release of our code and models—fully address the reviewer's concerns and strongly support the claims made in our paper.
>
> ------------------
> ## **Q4: The Whitening Transform's Complexity and Dataflow**
>
> We thank the reviewer for this question about the practical application of our whitening transform, `D`. Our method only modifies the computation paradigm of linear layers and does not alter the broader dataflow within FFN or Attention blocks. The core computation involves two steps, with the dataflow optimized based on the workload (e.g., decoding vs. prefill).
>
> **Scenario 1: Small Batch (e.g., Decoding, Memory-Bound)**
>
> In this common scenario, we apply the transform to the activation vector `A` to minimize computational overhead.
>
> 1. **Compute `A_tmp = DA`:** The transform `D` is applied to the small activation vector. As detailed in our paper, the structured nature of `D` (composed of diagonal and Kronecker product matrices) results in a low complexity of **`O(n√n)`**, where `n` is the activation dimension. This step incurs minimal overhead.
> 2. **Compute `W_q \* A_tmp`:** The main matrix-vector multiplication is then performed using the low-bit weights `W_q` and the transformed activation `A_tmp`. This is efficiently executed using optimized integer arithmetic kernels (e.g., from BitBLAS).
>
> This "transform-the-activation" approach ensures that the expensive `D` transform is only applied once per token to the small activation vector, making it highly efficient for decoding.
>
> **Scenario 2: Large Batch (e.g., Prefill, Compute-Bound)**
>
> In compute-bound scenarios with large batches, it is more efficient to pre-apply the transform to the weights.
>
> 1. **Compute `W_eff = W_qD`:** The effective full-precision weight matrix `W_eff` is reconstructed once. This dequantization step has a fixed cost per layer.
> 2. **Compute `W_eff \* A`:** The main computation then becomes a standard FP16 GEMM, which is highly optimized for large batch sizes on modern hardware.
>
> In both scenarios, the additional memory traffic is minimal, primarily consisting of loading the small, shared `D` matrix parameters once per layer. The detailed breakdown of FLOPs and storage overhead is provided in our response to **W1 & Q1**.

---

> ### Author Response · Authors · 2025-11-25
>
> -----------------------------
> ## **W5&Q6: Sensitivity to Calibration Data**
>
> We thank the reviewer for this important question about the sensitivity of our fine-tuning process to the calibration data. To address this, we conducted a series of ablation studies on Llama-2-7B, varying the calibration dataset's size, domain, and sequence length. The results are presented in **Table 9**.
>
> For these experiments, we used a `20/10` `M`-matrix and only performed the intra-block correction (no end-to-end fine-tuning) to isolate the effect of the calibration data.
>
> **Table 9: Ablation Study on Calibration Data for Llama-2-7B (2-bit).**
>
> | **Calibration Set**         | **Config. (Samples x SeqLen)** | **WikiText-2 PPL (↓)** | **C4 PPL (↓)** | **Avg. 0-shot Acc. (↑)** |
> | :-------------------------- | :----------------------------- | :--------------------- | :------------- | :----------------------- |
> | RedPajama (Small)           | 512 x 2048 (~1M tokens)        | 7.08                   | 8.66           | 60.03                    |
> | RedPajama (Medium)          | 1024 x 2048 (~2M tokens)       | 7.00                   | 8.59           | 60.55                    |
> | RedPajama (Large)           | 2048 x 2048 (~4M tokens)       | 6.96                   | 8.53           | 60.68                    |
> | **RedPajama (Default)**     | **4096 x 2048 (~8M tokens)**   | **6.97**               | **8.53**       | **60.70**                |
> | RedPajama (Short Seq)       | 4096 x 512 (~2M tokens)        | 6.98                   | 8.53           | 60.67                    |
> | WikiText-2 (Domain-Matched) | 2048 x 2048 (~4M tokens)       | **6.72**               | 8.65           | 60.24                    |
>
> Our findings from this study are twofold:
>
> 1. **Robustness to Data Size and Sequence Length:** Increasing the calibration data size from 1M to 8M tokens provides a noticeable benefit, but we observe **diminishing returns beyond ~4M tokens**. Similarly, using a shorter sequence length (512 vs. 2048) has a minimal impact on the final performance. For our main experiments, we chose 8M tokens to align with the settings used by other SOTA methods like AQLM and EfficientQAT, ensuring a fair comparison.
> 2. **Impact of Domain Shift:** As expected, calibrating on a domain-matched dataset (WikiText-2) yields the best perplexity on that specific domain (6.72 PPL). However, this comes at the cost of slightly degraded performance on the out-of-domain C4 dataset and zero-shot tasks. Using a general-purpose corpus like RedPajama provides a more balanced and robust performance across all benchmarks.
>
> --------------------
> ## **Q5: Regarding Fractional Bit-Widths, Storage, and Pareto Plots**
>
> We thank the reviewer for these questions. Here are the clarifications on our fractional bit-widths.
>
> ### **1) Mapping and True Model Size:**
>
> Our fractional bit-width is determined by the ratio of the lifted dimension to the subspace dimension. For example, "2.3-bit" uses a `16/7` `M` matrix (`16/7 ≈ 2.29`). This naturally translates to the final model size, as we store the 1-bit indices for the full lifted dimension. The tables below report the true storage cost for the transformer blocks, including all weights and transforms.
>
> | **Model & Config.**  | **Llama2-13B** | **Qwen2.5-3B** | **Qwen2.5-14B** |
> | :------------------- | :------------- | :------------- | :-------------- |
> | **~2.0-bit (20/10)** | 2.96 GB        | 0.65 GB        | 3.08 GB         |
> | **~2.3-bit (16/7)**  | 3.40 GB        | 0.74 GB        | 3.54 GB         |
> | **~2.7-bit (19/7)**  | 3.99 GB        | 0.87 GB        | 4.15 GB         |
> ***Only including weights in Transformer blocks***
>
> ### **2) Pareto Plots with Error Bars:**
>
> Regarding error bars, we must respectfully note that running each experiment multiple times to generate them is **beyond our current computational resources**. We acknowledge this limitation and believe our consistent results across many models still provide strong evidence.

---

> ### Author Response · Authors · 2025-11-25
>
> --------------------
> ## **Q7: Kernel details**
>
> We thank the reviewer for this detailed question about our kernel implementation.
>
> First, we must respectfully clarify our hardware limitations. We were only able to conduct stable end-to-end throughput benchmarks on a consumer **NVIDIA RTX 4090D GPU**. Unfortunately, we do not have access to A100 or H100 GPUs for further profiling. Our A800 server exhibited highly unstable performance during these specific latency tests, making the results unreliable. We have presented our complete 4090D benchmark results in our response to **W1 & Q1**.
>
> To provide full transparency, we have **open-sourced all benchmarking code and kernel implementations** at: `[Your Anonymized URL Here]`. Below, we describe the implementation details.
>
> **1. Triton Kernel for the `D` Transform:**
>
> Our whitening transform `A_tmp = DA` is implemented in Triton. Due to the very low computational load of this transform, we found through auto-tuning that a single Triton block is sufficient for optimal performance, minimizing kernel launch overhead. Within this single block, all four operations (`s1`, `P1`, `P2`, `s2`) are fused.
>
> For example, assuming an activation block `a` of shape `[N, M]`, `P1` of `[N,N]`, `s1` of `[N,M]`, `P2` of `[M,K]`, etc., the computation within the Triton kernel can be conceptually expressed as: `A_tmp = (P1 @ (a * s1) @ P2) * s2`.
>
> **2. BitBLAS Kernel for the Main GEMV:**
>
> For the main matrix-vector multiplication, `W_q * A_tmp`, we directly use the standard integer GEMV kernels provided by the official **BitBLAS** library. No modifications are required, showcasing the ease of integrating LiftUQ with existing, highly-optimized libraries.
>
> Interestingly, during our end-to-end evaluation (presented in W1&Q1), we observed that a simple PyTorch implementation yielded higher end-to-end throughput than our Triton kernel. We hypothesize this is due to the kernel launch overhead outweighing the small computational gains from Triton fusion. Therefore, the **final reported end-to-end results use a PyTorch + BitBLAS pipeline**, which represents a practical and highly efficient implementation.

---

> ### Author Response · Authors · 2025-11-27
>
> Dear Reviewer, We sincerely thank you for your insightful comments and valuable feedback. We are very grateful for the time and effort you dedicated to reviewing our manuscript. Your suggestions have been instrumental in helping us substantially improve the quality and clarity of our paper.
>
> We have uploaded a revised version of our manuscript. The major changes are summarized below:
>
> 1. **Discussion of M-Matrix Dimension and layer-reuse.**
> 2. **Decoding Flow and Complexity.**
> 3. **Component Ablation Study.**
> 4. **Calibration Data Analysis.**
> 5. **Detailed Fractional Bit-width Explanation.**
> 6. **Expanded Experimental Evaluation.**
>
> All revisions are **marked in blue** for your convenience. We believe these revisions have significantly strengthened our paper and have thoroughly addressed the points you raised. We look forward to your feedback.

---

### Author Response · Authors · 2025-12-03
**Authors' Final Summary for AC**

We thank all reviewers for their time and effort in evaluating our manuscript. We appreciate the thoughtful feedback, which has helped us to further refine and strengthen our work. In light of the suspension of the standard rebuttal process, we would like to provide a concise summary of our work's key contributions and the extensive new results we have prepared in response to the reviewers' valuable comments. We have revised our manuscript accordingly, with all new additions marked in blue.

------

## 1. Summary of Key Contributions & Novelty

As recognized by multiple reviewers, LiftUQ introduces a novel and practical paradigm for extreme low-bit quantization, built upon several key innovations:

1. **A New, Hardware-Aware Principle for Quantization:** LiftUQ is the first framework to demonstrate that expressive, non-uniform codebooks can be *procedurally generated* from a simple 1-bit uniform lattice in a lifted space via a learned linear projection. This elegant `lift-then-project` mechanism achieves the accuracy of Vector Quantization (VQ) while completely eliminating its inefficient, hardware-unfriendly lookup tables (LUTs), thus preserving the high efficiency of Uniform Quantization (UQ). *(Reviewers QuoP, d92k, kiJ4, xAD3)*
2. **Flexible and Pareto-Optimal Quantization:** Our method natively supports **fractional bit-widths** by encoding information in the dimensionality of the lifted space. This unique capability allows for fine-grained control over the model size, enabling the creation of a Pareto-optimal frontier where larger models, quantized with LiftUQ, can outperform smaller full-precision models under specific memory constraints. *(Reviewers QuoP, d92k)*
3. **A Principled and Efficient Three-Phase Pipeline:** LiftUQ employs a robust pipeline that decouples the problem into: (1) training a universal projection matrix `M` on Gaussian data, (2) learning a lightweight, layer-specific whitening transform `D`, and (3) performing lattice quantization with block-wise fine-tuning. This structured approach is both computationally tractable and highly effective, as validated by our strong empirical results. *(Reviewer QuoP)*

------

## 2. Summary of New Experiments and Analyses

In response to the reviewers' feedback, we have conducted a comprehensive suite of new experiments and analyses to fully validate our claims. We believe these additions have thoroughly addressed **ALL** raised concerns:

- **Comprehensive Ablation Study on Core Components:** We now provide a detailed ablation study that quantifies the contribution of each component of our whitening transform and the core lift-then-project mechanism. *(Reviewers QuoP, d92k, kiJ4, xAD3)*
- **Detailed Analysis of `M` Matrix Dimensions:** We present a new analysis on the impact of `M` matrix dimensions, showing the clear trade-off between theoretical accuracy (MSE on Gaussian), offline search cost, and end-to-end model performance (PPL). *(Reviewers QuoP, d92k, kiJ4, xAD3)*
- **Justification for the Shared Global `M`:** We provide a principled justification, supported by a computational cost analysis (a per-layer `M` would be >100 GPU-hours on 7B LLM), for our design choice of using a shared global `M` with a layer-specific whitening transform `D`. *(Reviewers QuoP, kiJ4, xAD3)*
- **Sensitivity Analysis of Calibration Data:** We now include a new study on the sensitivity to the calibration dataset's size, domain, and sequence length. *(Reviewers QuoP, d92k)*
- **In-depth Comparison with QTIP:** We have added a new section providing a head-to-head comparison with QTIP, demonstrating that LiftUQ achieves comparable or better accuracy with significantly higher, hardware-friendly inference throughput. *(Reviewers kiJ4, xAD3)*
- **Detailed Decoding Cost Model and Expanded Benchmarks:** We provide a formal decoding cost model, a detailed breakdown of offline quantization time, and expanded end-to-end throughput results. *(Reviewers QuoP, d92k, kiJ4, xAD3)*
- **Validation on Diverse Architectures and Benchmarks:** We have expanded our evaluation to include MoE (Mixtral 8x7B) and instruction-tuned models (Qwen-Instruct), as well as the comprehensive MMLU benchmark. *(Reviewers QuoP, d92k)*
- **Clarification on Differences from Prior Works:** We have added detailed discussions clarifying the fundamental differences between LiftUQ and other methods like RabitQ and BCQ, highlighting our unique contribution. *(Reviewers d92k, xAD3)*
- **Analysis of Theoretical Limits:** We provide a clearer explanation for our "0.15-bit information gap" estimate and its practical utility in guiding deployment choices. *(Reviewer kiJ4)*

All modifications are clearly marked in blue in the revised manuscript. We are confident that our work introduces a significant and practical new paradigm for extreme low-bit quantization, and we believe our extensive revisions have fully addressed all questions raised. We hope you will find our contributions valuable for the ICLR community.

---

### Meta-Review · Area_Chair_Jgd7 · 2025-12-28

**Summary:**

This paper proposes "LiftUQ," a method to compress Large Language Models (LLMs) to extreme low bits (like 2-bit) using a "lift-then-project" approach. The goal is to combine the accuracy of Vector Quantization with the speed of Uniform Quantization. However, the initial evaluation from the reviewers was very low. Reviewers QuoP, kiJ4, and xAD3 gave a score of 2 (Reject), and Reviewer d92k gave a score of 4 (Borderline Reject). They raised serious concerns about missing baselines, insufficient experiments, and unclear computational costs. Although the authors provided a very detailed Rebuttal and new results to answer these questions, the initial scores were so low that it is difficult to expect the paper to cross the Acceptance criteria, even if the reviewers raise their scores positively.

**Reviewer Concerns:**

**Addressed**
 - Missing Baselines: Reviewers d92k, kiJ4, and xAD3 pointed out the lack of comparisons with state-of-the-art methods like QTIP and BCQ. The authors added detailed comparisons and discussions for both QTIP and BCQ in the revision.
 - Ablation Studies: Reviewers asked for an analysis of the "M matrix" dimensions and the whitening transform components. The authors provided comprehensive ablation studies showing the impact of these components.
 - Decoding Cost and Speed: Reviewer QuoP questioned the decoding complexity and requested end-to-end benchmarks. The authors provided a formal cost model and throughput comparisons against methods like QuIP# and AQLM.
 - Experimental Scope: Reviewers suggested testing on more diverse models. The authors added results for Mixtral (MoE) and Qwen-Instruct models.

**Outstanding**
 - Paradigm Shift Claim: Reviewer xAD3 questioned whether this method is truly a "new paradigm" or just an incremental improvement. While the authors argued for their novelty, this is a subjective point that may still concern the reviewer.
 - Practicality of Offline Search: Reviewer QuoP was concerned about the "nearest-neighbor search" bottleneck during quantization. The authors explained a heuristic search to solve this, but the high offline cost (~100 hours for 70B models) remains a significant factor.

**Reviewer Scores:**

- Reviewer d92k: This reviewer explicitly stated they are "more inclined to support the publication" after the rebuttal. I believe their score would likely change from 4 to 6.
 - Reviewer QuoP: The authors answered the specific questions about decoding costs and kernels. However, given the initial score of 2, the score would likely only improve from 2 to 4 at most.
 - Reviewer kiJ4: The authors addressed the comparison with QTIP and the "0.15-bit gap" question. The score might improve from 2 to 4 at most.
 - Reviewer xAD3: The authors provided the requested comparisons and ablations. However, since the reviewer doubted the core novelty, the score might not be changed.

---

### Decision · Program_Chairs · 2026-01-26

Reject